# Structural barriers control the spatial extent of slow earthquake slip

**Takeshi Akuhara** [1] ✉, **Kazuya Shiraishi** [2], **Takeshi Tsuji** [3], **Yusuke Yamashita**[4], **Hiroko Sugioka** [5], **Atikul Haque Farazi** [6], **Shukei Ohyanagi**[7], **Yoshihiro Ito** [8], **Ryuta Arai** [2], **Eiichiro Araki**[2], **Gou Fujie** [2], **Yasuyuki Nakamura** [2], **Takashi Tonegawa** [2], **Ryosuke Azuma**[9], **Ryota Hino**[9], **Kimihiro Mochizuki** [1], **Shunsuke Takemura** [1], **Tomoaki Yamada** [1] & **Masanao Shinohara** [1]

Recent seismological and geodetic observations have revealed shallow slow earthquakes on subduction megathrusts, offering an opportunity to explore the factors controlling megathrust slip. Identifying these controlling factors helps constrain the potential updip extent of future megathrust ruptures. Here, we investigate the factors controlling the spatial extent of shallow slow earthquakes by integrating tremor locations with uncertainty estimates, detailed bathymetry data, and multichannel seismic data for the 2020–2021 slow earthquake sequence in the Kumano-nada region of the Nankai Trough. Tremor probability maps reveal that the updip, downdip, and lateral limits of tremor activity coincide with distinct structural boundaries. These include the kink or branching of the décollement, a deep-seated strike-slip fault, and the inner–outer wedge boundary. These structures likely act as geometrical or mechanical barriers, further influenced by variations in pore-fluid pressure and by the subducted Paleo–Zenisu ridge. The results indicate that megathrust geometry, material properties of the overriding prism, fluid distribution, and ridge subduction jointly govern the spatial extent of slow earthquakes, emphasizing the need to account for these factors in assessing the potential extent of future megathrust ruptures.

Recent seismological and geodetic observations have increasingly detected slow earthquakes located on the shallow portions of subduction zone megathrust faults[1–6]. Because they occur more frequently than megathrust earthquakes, shallow slow earthquakes offer important insights into megathrust slip behavior[7–10]. Specifically, delineating the spatial extent of shallow slow earthquakes and interpreting them as indicators of slip tendency along a megathrust may help identify factors that impede the slip propagation of future megathrust events. The exploration of such factors is ongoing. For example, subducted seamounts[11] and megathrust curvature[12] have been reported to prevent rupture propagation. Understanding whether slip can propagate to the trench is also crucial for assessing tsunami hazards[13].

Considerable efforts have been made to elucidate the relationship between slow earthquakes and subsurface structures, primarily to understand the generation mechanisms of slow earthquakes. These

[1]Earthquake Research Institute, The University of Tokyo, 1-1-1 Yayoi, Bunkyo-ku, Tokyo, Japan. [2]Research Institute for Marine Geodynamics, Japan Agency for Marine-Earth Science and Technology, 2-15 Natsushima, Yokosuka, Kanagawa, Japan. [3]School of Engineering, The University of Tokyo, 7-3-1 Hongo, Bunkyo-ku, Tokyo, Japan. [4]Faculty of Humanities, Miyazaki Municipal University, 1-1-2 Funatsuka, Miyazaki-shi, Miyazaki, Japan. [5]Graduate School of Science, Kobe University, 1-1 Rokkodai-cho, Nada-ku, Kobe, Japan. [6]Department of Geology and Mining, University of Barishal, Kornokathi, Barishal, Bangladesh. [7]Graduate school of science, Kyoto University, Gokasho, Uji, Kyoto, Japan. [8]Disaster Prevention Research Institute, Kyoto University, Gokasho, Uji, Kyoto, Japan. [9]Graduate School of Science, Tohoku University, 6-6 Aza-Aoba, Aramaki, Aoba-ku, Sendai, Miyagi, Japan. ✉e-mail: akuhara@eri.u-tokyo.ac.jp

studies have identified structural characteristics that may be associated with slow earthquakes, such as fluid-rich sediments[14–16], subducted seamounts[17–19], smaller-scale topographic irregularities[20], and material heterogeneities[21]. However, these studies have revealed only coarse spatial correlations between the structural factors and the main characteristics of slow earthquake distributions, such as their spatial centroids. Structural causes that delineate the edge of slow earthquake distributions remain elusive.

Slip propagation during slow earthquake episodes, which often span several months, can be traced in detail by analyzing the spatio-temporal evolution of tectonic tremor swarms, each comprising numerous tremors lasting from tens to hundreds of seconds[22,23]. This approach is based on the working hypothesis that a slow slip event (SSE) generates tectonic tremors as its migration front encounters fine-scale heterogeneities on the fault[24,25]. This hypothesis is supported by the frequent observation of spatiotemporal synchronization between tremors and the SSE slip front[23], although some SSEs have been reported to occur without accompanying tremor activity[26]. In recent years, such synchronization has also been observed between shallow tremors and SSEs in the Kumano-nada region of southwestern Japan, the focus region of this study[27,28].

However, precisely locating tectonic tremors remains challenging because the onset of their seismic signals is often unclear. In addition, for offshore locations, the laterally heterogeneous structures of accretionary prisms can introduce bias in the estimated locations of tectonic tremors[29]. To tackle these challenges, a recent study[30] developed a Bayesian inversion method that uses seismic amplitudes as input data to estimate tremor locations. This approach provides better constraints than conventional time difference data[31]. This method also accounts for laterally heterogeneous structures by solving for structural parameters that compensate for these variations. Furthermore, the Bayesian framework performs probabilistic sampling of tremor locations, providing formal uncertainty estimates and enabling representation of the source region as a probability map (see Methods).

The previous study[30] applied this method to tremors occurring in the Kumano-nada region of the Nankai Trough subduction zone between December 2020 and February 2021 (Fig. 1). The results showed that median location errors across all events are 1.1 km in the east–west direction and 1.2 km in the north–south direction. These errors represent the half-widths of the 95% confidence intervals (Supplementary Fig. 1). Notably, the 2020–2021 tremor episode was the most extensive activity recorded by the Dense Oceanfloor Network system for Earthquakes and Tsunamis (DONET)[32,33], a seafloor-cabled network deployed in this region[9,34]. By comparison, the previous tremor episode in April 2016 exhibited a source region approximately half as wide[34]. Seismic signals during the 2020–2021 tremor episode were recorded not only by the DONET stations but also by temporarily deployed ocean-bottom seismometers[30]. The unusually intense activity, together with the enhanced observation network, provided an unprecedented dataset for the present study.

In this study, we combine tremor locations with multichannel seismic (MCS) images acquired along survey lines covering the Kumano-nada region to explore the structural factors that define the spatial extent of slow earthquakes. The location uncertainties of tremors, approximately 1 km, enable meaningful comparisons with MCS images, which typically have spatial resolutions on the order of 100 m. To supplement the two-dimensional nature of the MCS data, we also examine bathymetric features, which closely approximate subsurface structural characteristics[35]. Together, these datasets show that the distribution of tectonic tremors is correlated with structural boundaries, indicating the structural controls on fault-slip behavior.

## Results

### Structural features correlated to tremor distribution

The Nankai Trough subduction zone, a typical accretionary system, is characterized by deformation features on the bathymetry, notably fold-and-thrust structures (Fig. 2). These structures are primarily concentrated within an outer wedge located trenchward of the megasplay fault. The outer wedge is divided into two geological zones:

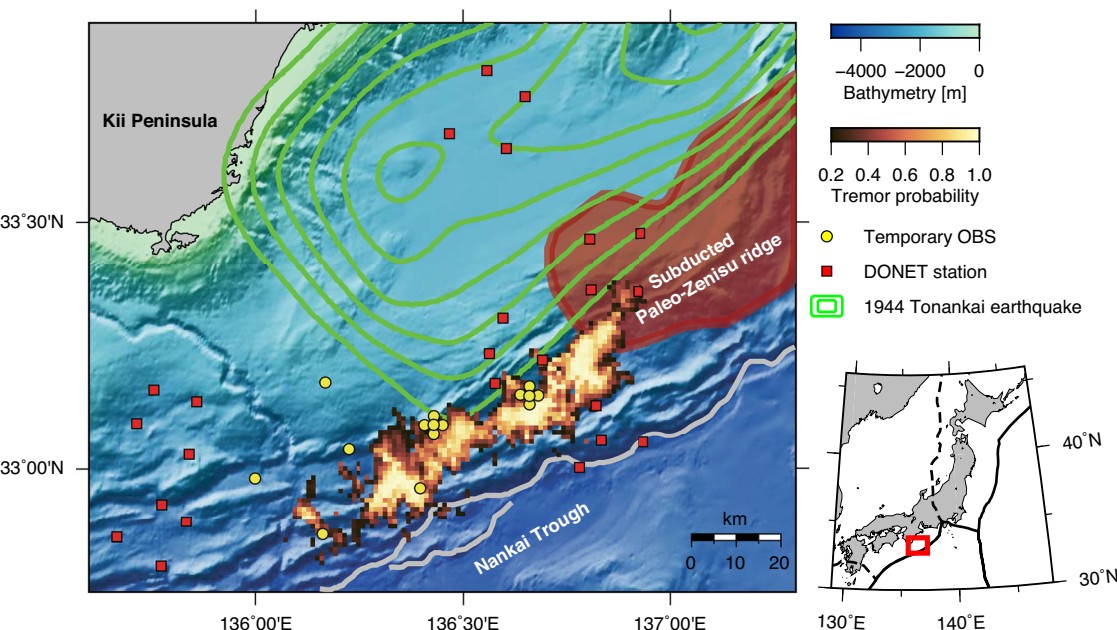

**Fig. 1 | Tectonic setting of the study region.** Tremor probability is shown by the brownish/cream heatmap (probabilities lower than 0.2 are muted for visibility). Circles and rectangles represent the temporarily deployed ocean-bottom seismometers and DONET stations, respectively, used to locate the tremors. Green contours depict the slip distribution of the 1944 Tonankai earthquake at 0.5-m intervals, reproduced from ref. 60 with permission from SNCSC. The red-shaded area delineates the subducted Paleo–Zenisu ridge, reproduced from ref. 40 with permission from Elsevier.

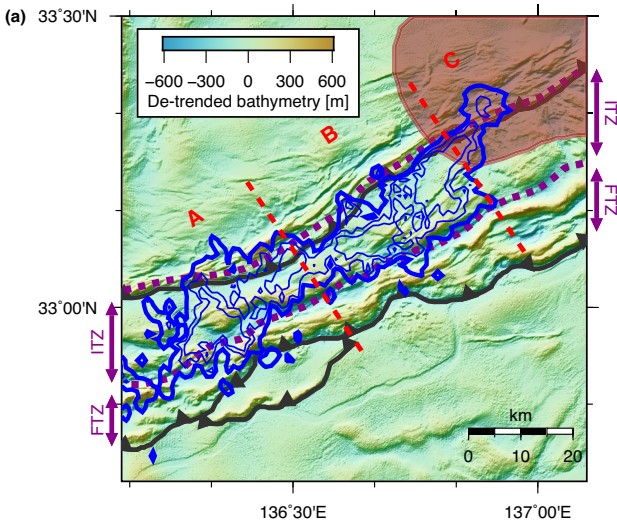

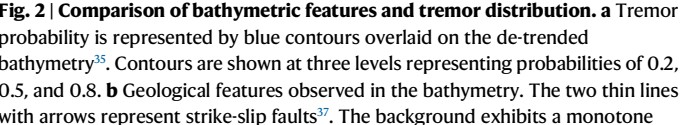

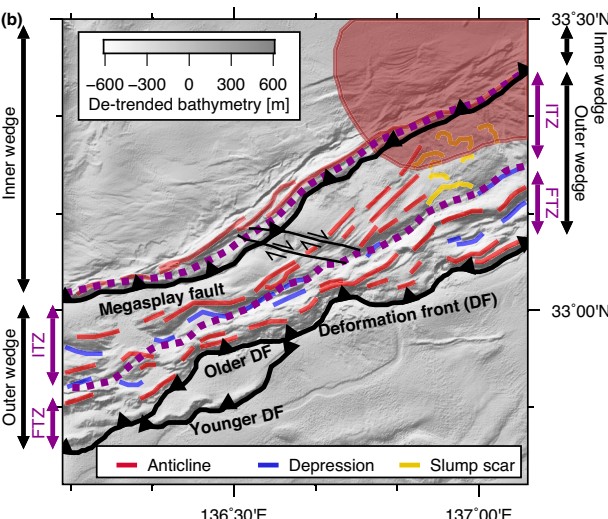

**Fig. 2 | Comparison of bathymetric features and tremor distribution. a** Tremor probability is represented by blue contours overlaid on the de-trended bathymetry[35]. Contours are shown at three levels representing probabilities of 0.2, 0.5, and 0.8. **b** Geological features observed in the bathymetry. The two thin lines with arrows represent strike-slip faults[37]. The background exhibits a monotone representation of the same de-trended bathymetry as shown in (**a**). The purple dashed lines indicate boundaries between the ITZ and FTZ, and between the inner and outer wedges. The red-shaded area outlines the subducted Paleo–Zenisu ridge, reproduced from ref. 40 with permission from Elsevier. ITZ imbricate thrust zone, FTZ frontal thrust zone, DF deformation front.

the landward, older imbricate thrust zone (ITZ) and the trenchward, younger frontal thrust zone (FTZ)[36]. These zones are roughly demarcated by a bathymetric depression running parallel to the trench[36]. In addition, two subparallel right-lateral strike-slip faults are located in the ITZ[35,37,38] (two thin black lines with arrows in Fig. 2b). These faults are inferred to be sufficiently deep-rooted to intersect the underlying décollement (i.e., the megathrust)[37]. The deformation front is also evident in the bathymetry map. In the western region (region A in Fig. 2a), a younger deformation front has been developing[36,39], while the older one remains active in other areas.

The 2020–2021 tremor episode probability map indicates that the tremor sources are mostly confined within the ITZ. The downdip limit of tremors correlates with the boundary between the inner and outer wedges. For the central region (region B in Fig. 2a), the updip limit of tremors mostly coincides with the ITZ–FTZ boundary. This close correspondence between tremors and geological zoning implies the presence of a structural factor that exerts primary control on the spatial extent of the tremors. While the absence of tremors is evident in the FTZ of region B, in the western region (region A), the updip limit of the tremors partly crosses the ITZ–FTZ boundary and approaches the old deformation front.

Fold-and-thrust features are poorly expressed in the eastern region (region C), east of the tremor distribution. Instead, the bathymetry is dominated by slump scars[35,36,38] commonly attributed to the subduction of the Paleo–Zenisu ridge[40]. Intriguingly, the transition of bathymetric features from region B, characterized by fold-and-thrust geometry, to region C, lacking such features, aligns with the eastern boundary of the tremor distribution. Furthermore, in region C, tremors occur further landward beneath the inner wedge, highlighting the distinct tectonic setting of region C, compared to regions A and B.

To further explore the correspondence between the tremor distribution and bathymetric features, we analyze and interpret MCS data from seven seismic reflection profiles traversing the tremor zone (Fig. 3 and Supplementary Fig. 2–7). Profiles A1–A2 are located in region A, B1–B3 in region B, and C1–C2 in region C. The most prominent features observed in the resulting seismic images are reflections from the top of the subducting oceanic crust and décollement, which are subparallel within the ITZ. The overriding prism hosts numerous faults, mainly reverse faults, including a few that may be strike-slip faults[37].

In regions B and C, the décollement undergoes a sudden change in dip angle and extends upward to the deformation front (profiles B1–B3 and C1–C2 in Fig. 3). Hereafter, we refer to this steepened portion of the décollement as the "steep thrust". The steep thrust extends westward into region A (profiles A1–A2). However, in this region, the décollement continues to farther seaward beneath the steep thrust and reaches the younger deformation front[39]. The steep thrust in region A is therefore interpreted as a frontal thrust branching off the décollement.

Comparison between tremor probability and seismic images shows that, in regions A and B, the updip limit of the tremors coincides with the kink and branching points of the décollement associated with the steep thrust (white stars in Fig. 3). This alignment suggests that these kink or branching points may have acted to impede slip propagation during the slow earthquake episode. Accordingly, the reliability of the estimated updip limit of tremors was evaluated through synthetic tests. This test confirmed that the station coverage was adequate for detecting and locating tremors that might have occurred farther seaward of the identified updip limit (Supplementary Fig. 8).

However, in region C, the updip limit of the tremors is distinctly separated from the kink points along survey lines C1 and C2 (Supplementary Fig. 4). This unique characteristic suggests the presence of another structural factor. As discussed later, we attribute this to the subducted Paleo-Zenisu ridge.

The downward continuation of the décollement beneath the inner wedge is difficult to delineate in the seismic reflection profiles because of its weak and relatively incoherent reflections (Fig. 3). Nevertheless, the top of the oceanic crust can still be identified, providing a reasonable proxy for the décollement, given that the separation between the two interfaces remains nearly constant.

The top of the oceanic crust generally dips more steeply beneath the inner wedge than beneath the ITZ. However, the manner in which the dip angle increases with subduction varies among the profiles across the study area. For example, along profile B3, the dip angle increases sharply near the boundary, whereas along profile B2, it remains nearly constant and gradually steepens with subduction.

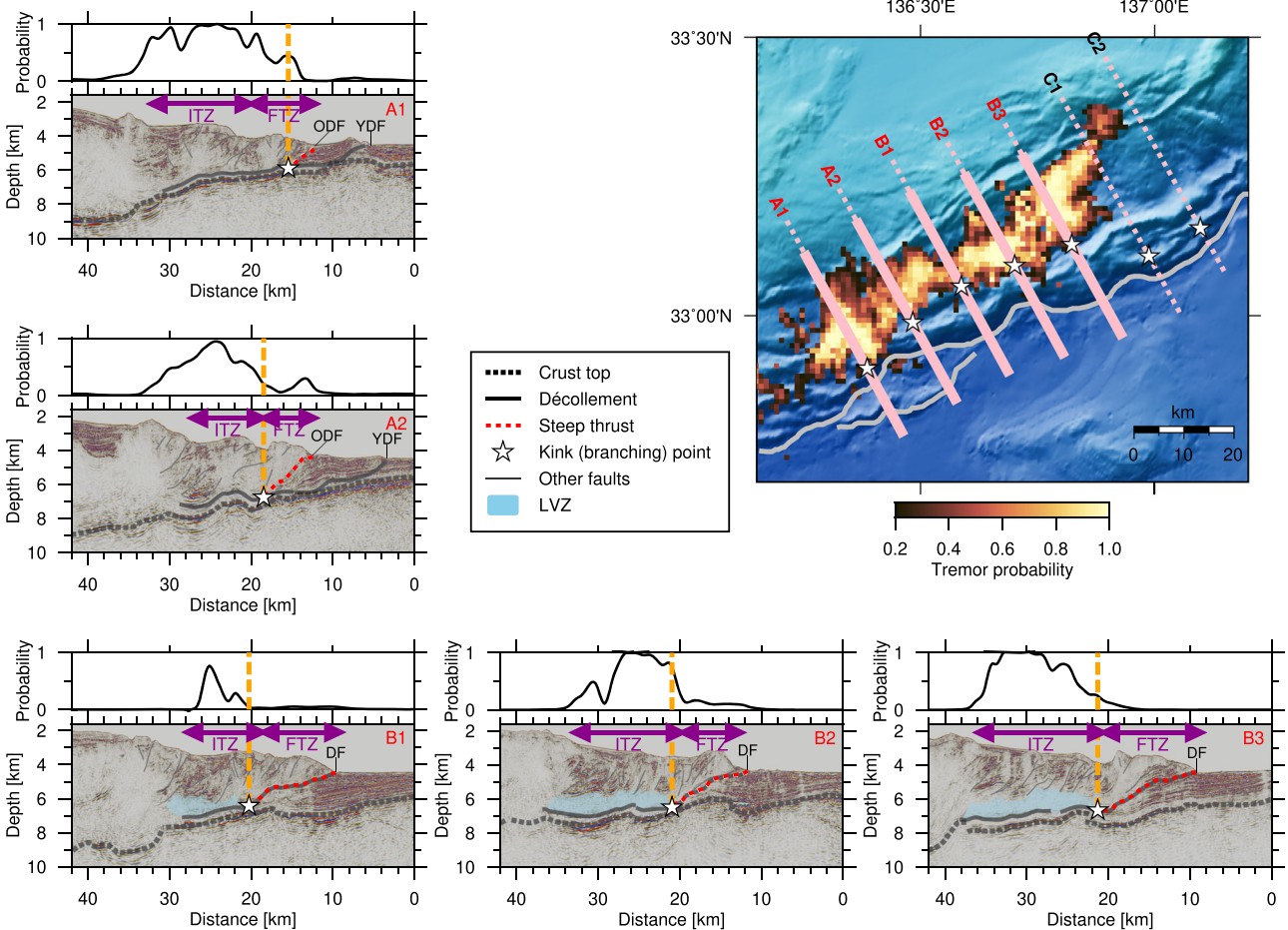

**Fig. 3 | Comparison of seismic reflection sections and tremor distribution.** In each panel, the top panel shows the tremor probability along the survey line, and the bottom panel shows the seismic reflection image along the survey line. The thick solid and dashed lines represent the décollement and the top of the subducting oceanic crust, respectively. The thick red lines represent the frontal thrust. Blue-shaded area indicates a zone of transparent seismic facies interpreted as a low-velocity zone (LVZ). The white stars represent the branching point of the frontal thrust on the décollement. The inset map shows the tremor probability as shown in Fig. 1, and the locations of the branching points are indicated by the white stars. The pink lines represent the locations of the reflection sections, where the thick, solid portions represent the locations of the reflection sections shown here. The complete sections are shown in Supplementary Figs. 2–7. DF deformation front, YDF younger deformation front, ODF older deformation front, ITZ imbricate thrust zone, FTZ frontal thrust zone.

Therefore, the downdip limit of tremors does not correlate with the subduction geometry.

What appears to correlate more strongly with the downdip limit of tremors is the seismic characteristics of the overriding prism. In the ITZ, where tremors occur, the prism exhibits high reflectivity and contains numerous faults. By contrast, within the inner wedge, the prism is characterized by transparent seismic facies, except in the uppermost ~1 km beneath the seafloor, corresponding to the basin-fill sediments of the Kumano Basin.

### Geometrical and hydrological barriers controlling the updip limit of slow earthquake slip

Under the hypothesis that the slip front of SSEs generates tremors along the décollement[24,25], the spatiotemporal evolution of tectonic tremors effectively approximates the slip propagation of ongoing SSEs along the megathrust[23]. Within this framework, the findings of the current study indicate that the along-dip propagation of SSEs is impeded at the kink or branching points. In the following, we explore the potential mechanisms for this impediment, and the limitations of this framework will be discussed later.

The geometry of the megathrust can be a primary factor that impedes slip propagation at the kink in region B. Under stress conditions inferred from two-dimensional non-cohesive Coulomb wedge

theory[41], the steep thrust experiences higher normal stress than the gently dipping section of the décollement (see Methods). In addition, a laboratory friction experiment suggests that the décollement exhibits a higher friction coefficient at this steep thrust than at the landward flat portion[42]. Consequently, the shear strength, represented by the product of normal stress and friction coefficient, is greater at the FTZ than at the ITZ. The steep thrust acts as a geometrical barrier when the dynamically perturbed shear stress at the slip front fails to exceed this increased shear strength[12].

Similar implications have also been reported by a previous investigation[43]. That study calculated the slip tendency (the ratio of shear stress to normal stress) along the décollement within the relatively narrow area covered by a three-dimensional reflection seismic survey in the Kumano-nada region. By fully accounting for the three-dimensional geometry of the décollement, the study demonstrated that the abrupt increase in the dip at the kink indeed decreased the slip tendency of the megathrust, consistent with our findings.

It remains unclear why slip terminates at the branching point along profiles A1 and A2 in region A, despite the continuation of the décollement further seaward. A possible explanation is that this segment of the décollement undergoes stable sliding. Notably, the reflection pattern in the hanging wall is different across the steep thrust. A stratified pattern observed at the trench side of the steep

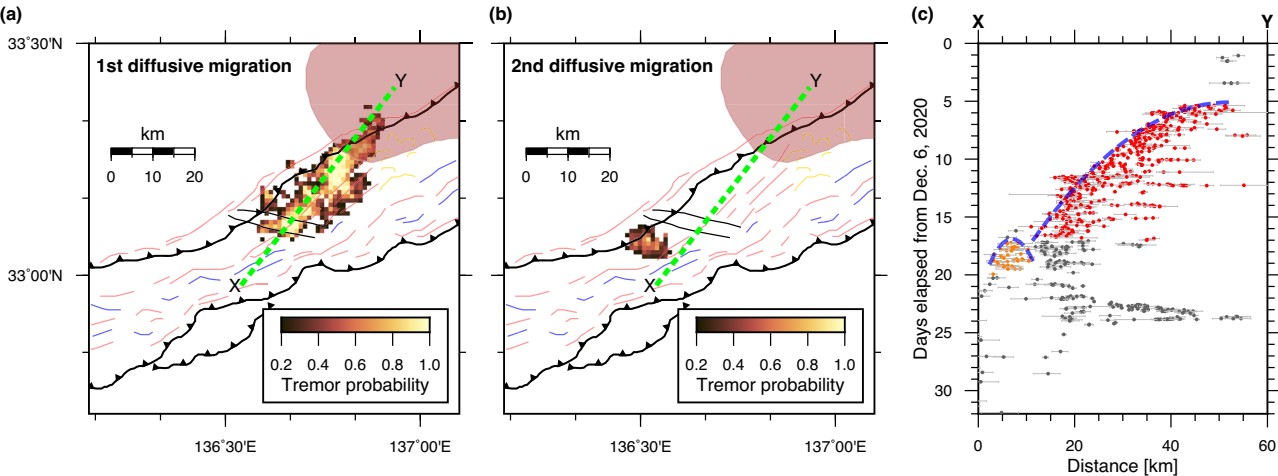

**Fig. 4 | Diffusive migration of tremors impeded by the strike-slip faults.** Tremor probability calculated from tremors during **a** first and **b** second diffusive migrations are shown by the heat map. Geological interpretation is depicted using the same notation as that used in Fig. 2b, including the strike-slip faults represented by subparallel black lines. The position of the subducted Paleo–Zenisu ridge (red-shaded area) is based on ref. 40 and is reproduced with permission from Elsevier. **c** Time–distance plot of tectonic tremors along the green dashed line shown in (**a**) and (**b**). The dots represent tremors, where the red and orange dots are those used to calculate the probability maps shown in (**a**) and (**b**), respectively, while the gray dots indicate tremors that are not included in either calculation. The error bars in (**c**) depict the range of the 95% confidence interval. The blue dashed parabola delineates the diffusive migrations of the tremors.

thrust is absent on the landward side (Fig. 3). The less deformed stratification at the trench side appears to be consistent with the idea of stable sliding. Even if this is not the case, the contrast in the reflection pattern likely reflects differences in physical, material, and/or hydrological properties, which may result in distinct slip behaviors. A more detailed investigation, including core sampling and numerical modeling, would be valuable for understanding the slip behavior near the trench for this region.

Pore fluid pressure also influences effective normal stress and hence slip behaviors[44]. Although its spatial distribution is not well constrained, the potential role of pore fluid pressure in the study region is worth discussing here. In region B, along survey lines B1–3, a ~1–2 km thick zone of transparent seismic facies is observed immediately above the décollement within the ITZ (Fig. 3 and Supplementary Fig. 3). Independent seismological studies using near-vertical reflection[45], wide-angle refraction[16], and teleseismic converted waves[46] have revealed low seismic velocities for this structure. Because of its high Poisson's ratio, this low-velocity zone (LVZ) has been interpreted as a zone of high pore fluid pressure[46]. The fluid is thought to originate mainly from disequilibrium compaction of sediments, and high pore fluid pressure is interpreted to be sustained by a low permeability of the sediment[44].

While it is challenging to determine the exact spatial extent of this LVZ from the two-dimensional reflection image alone, its updip limit seems to be near the kink point, and the LVZ is seemingly absent above the steep thrust (Supplementary Fig. 3). Independent studies have similarly shown no supportive evidence for an LVZ above the steep thrust[37,45,46]. Moreover, due to the high permeability along faults, the steep thrust probably acts as a pathway that leaks fluid from the LVZ to the seafloor. Under this interpretation, the steep thrust likely has lower pore fluid pressure, implying higher shear strength, and help the geometrical barrier terminate slip propagation at the kink point.

In region A, the presence of such an LVZ is not evident in the reflection image from this study. However, a previous study[47]. identified reflectors with negative polarities slightly above the décollement, which may indicate the top of the LVZ (Supplementary Fig. 2). Further investigation into the properties of the LVZ, including its spatial extent and associated pore fluid pressure, is necessary to better understand its role in fault slip propagation.

## Geometrical barrier controlling the along-strike propagation of slow earthquake slip

Another geometrical barrier is found in the along-strike slip propagation. As shown in Fig. 4, the 2020–2021 tremor episode began at the eastern end of the tremor distribution, subsequently migrating southwestward by ~40 km. The migration distance is proportional to the square root of time, as illustrated by the parabolic trajectory in the time–distance plot (Fig. 4c). Such a parabolic, or diffusive, migration pattern can be explained by a heterogeneous fault surface consisting of a ductile background and brittle patches[48], where tremors occur at the brittle patches behind the SSE migration front. Notably, this diffusive migration ceased when the migration front encountered a strike-slip fault that intersects the ITZ (Fig. 4a, c). This strike-slip fault, which is inferred to intersect and offset the décollement[37], may act as a geometrical barrier, hindering slip propagation. Tremor activity in April 2016 also appears to have terminated when the migration front reached this strike-slip fault[47,49] (Supplementary Fig. 9).

For the 2020–2021 episode, however, a cluster of tremors occurred on the opposite side of the strike-slip fault almost simultaneously with the cessation of the initial migration (Fig. 4b). The time-distance plot for these clustered tremors shows a parabolic envelope that represents an isolated diffusive migration from the initial one. The first and second migrations are spatially separated by ~5 km. We interpret that the SSE driving the first migration was impeded at the strike-slip fault, and the resulting remote stress transfer, rather than direct slip propagation, triggered a second SSE ~5 km away. Given that the gap is considerably small compared to the slip area of the first migration, the stress transfer from the first migration appears sufficiently strong to activate the second one.

## Mechanical barrier controlling the downdip limit of slow earthquake slip

The downdip limit of the tremor distribution generally corresponds to the boundary between the inner and outer wedges in regions A and B (Fig. 2) Unlike the updip limit, however, the downdip limit does not necessarily appear to correlate with abrupt change in the dip angle. The large-scale tendency for the subduction angle to be greater beneath the inner wedge may exert some geometrical control on slip behavior in the same manner as discussed in the previous section, but the spatial correlation between the downdip limit of tremors and the wedge boundary

is much sharper than that between tremors and changes in dip angle, suggesting that the latter is unlikely the primary controlling factor.

Instead, a rigidity contrast within the overriding prism may be responsible for defining the downdip limit. Seismic velocity models suggest that the prism is more rigid in the inner wedge than in the outer wedge[50,51]. In particular, Fig. 10 of ref. 50 shows that the averaged P-wave velocity within ~3 km above the oceanic crust increases sharply across the boundary between the inner and outer wedges. This sharp velocity increase likely reflects a transition from a less consolidated, faulted prism in the outer wedge to a denser and more lithified prism in the inner wedge, implying a higher rigidity of the overlying material above the downdip limit. This contrast can also be inferred from the seismic reflection profiles, which show transparent and highly reflective facies in the inner and outer wedges, respectively. Such a rigidity contrast at the wedge boundary could act as a mechanical barrier to slip propagation, because a stiffer overlying prism requires greater shear stress to drive slip along the underlying interface.

In addition, multiple lines of evidence, including borehole breakouts[52], seismic anisotropy[53], and geomorphic signatures[35], reveal an abrupt change in the maximum horizontal stress direction across the boundary between the inner and outer wedges: the inner wedge undergoes trench-parallel compression near this boundary, whereas the outer wedge experiences trench-normal compression. The trench-normal compression at the inner wedge misaligns with the slip direction along the megathrust, thereby reducing the resolved shear stress available to drive slip propagation.

It should be noted, however, that the downdip limit for this episode may not necessarily correspond to the boundary between transient-slip and locked zones, as SSEs and tremors occurred further downdip of this boundary during earlier slow earthquake episodes prior to 2020–2021[54].

### Stress shadow induced by ridge subduction limiting slow earthquake propagation

Slump scars that obscure the fold-and-thrust structures in region C indicate the influence of the subducted Paleo–Zenisu ridge on deformation within the overriding prism (Fig. 2). Correspondingly, the reflection image along profile C2 shows relatively well-preserved horizontal sedimentary strata within the prism of the ITZ, on the trailing side of the subducted Paleo–Zenisu ridge (Supplementary Fig. 4). These strata may reflect the effects of a stress shadow caused by ridge subduction. Numerical experiments suggest that seamount or ridge subduction generates a wake of reduced tectonic loading and delayed sediment compaction on its trailing side, forming a stress-shadow zone characterized by lower effective stress and limited deformation[18,55]. We also find horizontal strata at the FTZ along the profile C1, which could be related to the subduction of the smaller, undocumented ridge (Supplementary Fig. 4).

Such stress shadows are thought to affect the generation of slow earthquakes through their influence on hydrological properties, specifically the porosity and permeability, of the prism. However, it remains controversial whether stress shadows either promote[18] or inhibit[55] slow earthquakes. The spatial correlation identified in this study between the non-tremorgenic zone and the less-deformed prism suggests that the stress shadow contributed to terminating slip propagation. In a typical accreted prism, progressive tectonic loading reduces porosity, leading to an increase in pore fluid pressure. In contrast, in stress-shadow zones on the trailing side of the ridge, porosity reduction occurs more gradually, leading to a slower buildup of pore fluid pressure[55]. The resulting relatively low pore-fluid-pressure zone may have prevented the 2020–2021 SSE from propagating into region C.

### Limitations and opportunities

The fundamental assumption of this study is that tremors always accompany SSEs. However, SSEs without associated tremors have been reported in several regions[26]. One possible scenario not captured by this study is that an SSE propagates into the steep thrust toward the deformation front but does not generate tremors there. In such a case, the steep thrust may be characterized by a relatively homogeneous fault surface with fewer brittle patches capable of generating tremors. In line with this scenario, borehole observations during minor SSE episodes other than the 2020–2021 episode have identified slip approximately 5 km landward of the deformation front, possibly along the step thrust, whereas tremor occurrence there during these episodes remains unclear due to undocumented error estimations[28].

In addition, because the present location method provides limited depth resolution, the exact vertical positions of tremor sources cannot be firmly constrained. It is therefore assumed that tremors occur along the décollement. In contrast, centroid moment tensor solutions for VLFEs during the 2020–2021 episode are well constrained: most events have focal depths of 6–8 km below sea level and low-angle (~6°) thrust mechanisms[56], suggesting that they occurred along or near the décollement. The overall agreement in the spatiotemporal evolution of tremors and VLFEs (Supplementary Fig. 10) further supports the interpretation that tremors, like VLFEs, occurred along or near the décollement.

Considering the uncertainties in the centroid depths of the VLFEs, which average about 1.3 km[56], we cannot rule out the possibility that tremors and VLFEs are generated within a heterogeneous mélange shear zone embedded in the LVZ above the décollement[21,47], whereas the SSE may occur along the décollement itself. Even in this scenario, the hypothesis that tremors are generated at the migration front of the SSE remains plausible, as the LVZ would deform in concert with, or be mechanically coupled to, the underlying décollement. The absence of tremors in the FTZ may then reflect the lack of such an LVZ above the steep thrust.

Exploring these possibilities requires further enhanced geodetic and seismic observation networks, as well as more sophisticated analysis methods for better constraining source locations. Such efforts are essential for advancing our understanding of the physical linkages among different modes of slow earthquakes, SSEs, VLFEs, and tremors, and the controlling factors that govern their generation.

### Implications for future megathrust earthquakes

Recent megathrust events, such as the 2004 Sumatra-Andaman and 2011 Tohoku earthquakes, have highlighted the significance of coseismic slip extending close to the trench, which produced devastating tsunamis[57,58]. Understanding the updip limit of megathrust events is therefore important for accurate hazard assessment. In the study region, the latest megathrust event occurred in 1944 (Fig. 1, Tonankai earthquake), although the inferred updip limit of this event varies among different slip models[59–61].

Based on the tremor distribution, three possible scenarios can be proposed for the updip limit of the future megathrust events in the study area. The first scenario is that the megathrust rupture stops at the downdip limit of the tremor zone, that is, the inner and outer wedge boundary. Such spatial complementarity between slow earthquakes and megathrust events has been observed in subduction zones worldwide[3,4,9,10], although the underlying mechanism remains debated. Another possibility is that the steep thrust acts as a geometrical barrier, forming the updip limit of the megathrust event. In the most devastating case, the rupture could extend all the way to the trench, as suggested by a study analyzing core samples collected by seafloor drilling[62]. While multiple numerical simulations have been conducted to assess the hazard of future earthquakes in this region[13,63], they have not fully accounted for the mechanical and geometrical barriers identified in this study. Future studies should evaluate the likelihood that these barriers can prevent dynamic rupture during a megathrust event and assess their influence on tsunami generation and height.

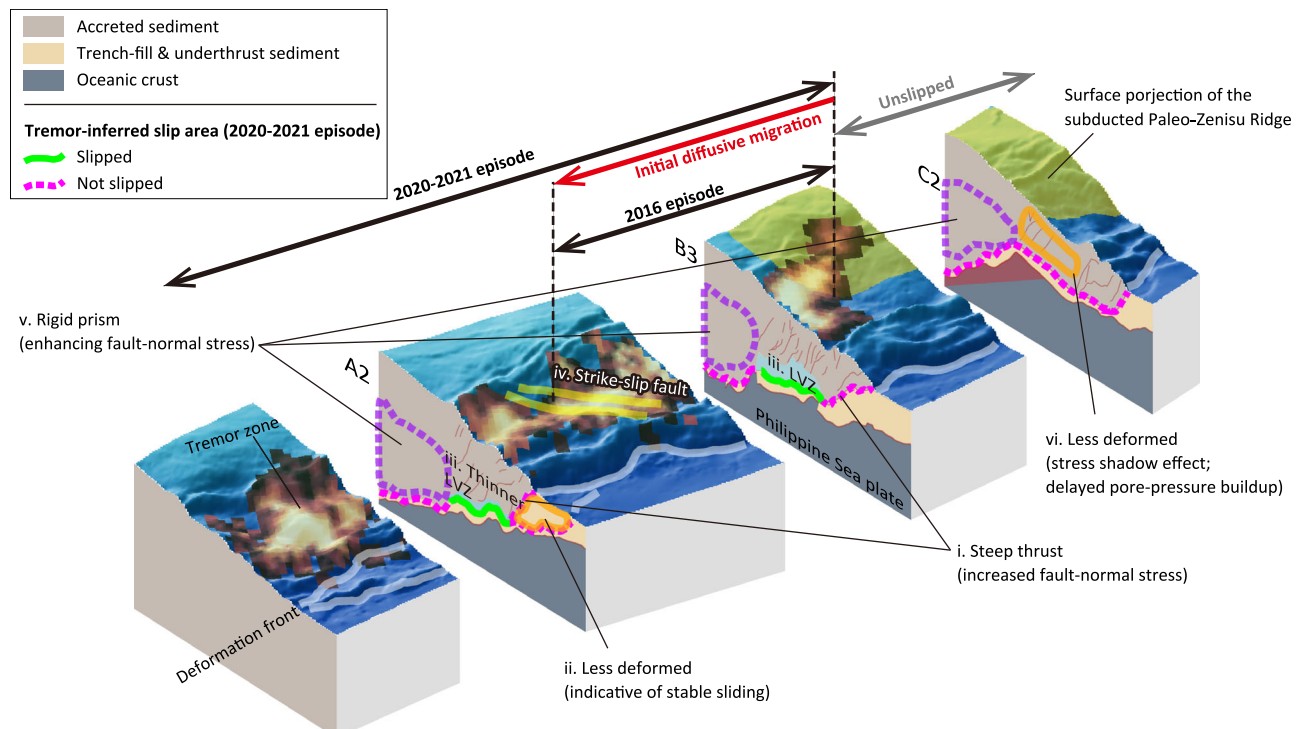

**Fig. 5 | Schematic illustration of our interpretation for the slip tendency on the megathrust.** Perspective view of the bathymetry overlaid by the tremor probability is sliced along survey lines A2, B3, and C2. Green solid and magenta dashed lines denote slip and unslipped area during the 2020–2021 episode, which is inferred from the tremor distribution. Roman numerals (i-vi) correspond to the key interpretations summarized in the main text: (i) steep thrust, (ii) less-deformed zone near the younger deformation front, (iii) LVZ above the décollement, (iv) strike-slip faults, (v) rigid prism, and (vi) less-deformed zone behind the subducted ridge. Arrows on top indicate the spatial extent of the 2016 and 2020–2021 episodes. The surface projection of the subducted Paleo–Zenisu ridge (yellow-shaded area) is based on ref. 40 and is reproduced with permission from Elsevier.

A global statistical compilation of megathrust earthquakes also emphasizes the critical role of the geometrical barriers in limiting rupture propagation[12]. Our study complements this global finding by providing a high-resolution comparison between tremor locations and subsurface structures. Extending the methodologies employed in this study to other regions could advance the understanding of megathrust dynamics. For example, near-trench slip of SSE (or associated tectonic tremors and VLFEs) has been reported for Hyuga-nada[6] and Costa Rica[3], but the spatial resolution is not sufficient for a detailed comparison to geological structures. This emphasizes the need for enhanced seismic monitoring and surveys that focus on shallow megathrusts beneath the ocean.

## Integrated controls on the spatial extent of fault slip
This study revealed the spatial correlation between tremor distribution in the Kumano-nada region and tectonic domains delineated by seafloor topography. Assuming that tremors occur at the migration front of SSEs, this spatial correlation suggests that the subsurface geological structures play a crucial role in controlling the spatial extent of megathrust slip.

The following mechanisms are proposed to impede the propagation of slow earthquake slip, as schematically summarized in Fig. 5:

i. The updip limit of tremors corresponds to the kink of the décollement toward the steep thrust in the central region (region B) and to the branching point where the steep thrust branches off from the décollement in the western region (region A). The steep thrust is considered to experience higher normal stress, acting as a geometrical barrier to slip propagation.

ii. In region A, the décollement continues horizontally seawards beyond the branching point. Less-deformed overriding material may indicate that this section undergoes stable sliding, leading to the absence of tremors and SSEs.

iii. The LVZ observed above the flat portion of the décollement represents a zone of high pore-fluid pressure, which appears to be absent above the steep thrust. This contrast in pore-fluid pressure may further contribute to the slip impediment at the megathrust kink.

iv. A deep-seated strike-slip fault inferred to cut across the décollement may hinder along-strike migration of SSEs.

v. The downdip limit of tremors corresponds to the boundary between the inner and outer wedges, where abrupt changes in material properties and stress orientation within the overriding prism likely define this limit.

vi. A stress shadow caused by the subducted Paleo-Zenisu ridge delays the buildup of pore-fluid pressure, preventing slip from propagating into its trailing side in the eastern region (region C).

These findings indicate that megathrust geometry, the physical properties of the overriding prism, fluid distribution, and ridge subduction jointly govern slip behaviors, emphasizing the importance of considering these factors when assessing the potential extent of future megathrust ruptures.

## Methods
### Tremor probability map
The Bayesian inversion analysis[30] used for locating tectonic tremors provides ensemble solutions sampled from posterior probabilities. For each specific event, we estimate the probability of its epicenter falling within a particular $1 \times 1$ km geographical cell using the following formula:

$$p_i(\mathbf{r_j}) = \frac{N_{r_j}}{M},$$

where $p_i(\mathbf{r_j})$ represents the posterior probability of the epicenter of the $i$th tremor, with respect to the spatial variable $\mathbf{r_j}$, which denotes the position of the $j$th cell; $N_{r_j}$ represents the count of the ensemble solutions that fall within the $j$th cell; and $M$ represents the total number of solutions.

We compute the probability of at least one tremor occurring within the corresponding cell by aggregating this probability across multiple events as follows:

$$p_{tremor}(\mathbf{r_j}) = 1 - \prod_{i=1}^{N_{evt}} \left[ 1 - p_i(\mathbf{r_j}) \right]$$

This formula exhibits two key attributes: reduced sensitivity to the number of tremors and decreased influence from highly uncertain events. Both features are advantageous for delineating the tremor-genic zone compared to conventional visualization using a scatter plot, as shown in Supplementary Fig. 1a. This study refers to this probability as tremor probability for simplicity.

## Tremor location reliability test

This study uses the tremor locations determined by a previous study[30] that employed a workflow including event prescreening and Bayesian inversion. The prescreening stage estimated the apparent S-wave velo-cities and attenuation strengths of seismic waves emitted by a tremor, assuming the tremor source was located beneath a station that recorded the highest signal amplitude within the seismic network. The resulting S-wave velocity and attenuation strength were used as quality control factors to retain events with good-quality data. However, this assump-tion may have increased the spatial variability of tremor detectability and potentially biased the spatial extent of the tremor distribution. To investigate this, we conducted the following synthetic test.

In the synthetic test, virtual sources were evenly distributed 7 km beneath sea level at 5 km horizontal intervals. Synthetic arrival time and amplitude were calculated for every source–receiver path. To simulate real conditions, random noise was introduced to the synthetic data: the arrival times were perturbed by random noise sampled from a zero-mean Gaussian distribution with a standard deviation of 4 s, and the amplitudes were multiplied by a random factor drawn from a Gaussian distribution with a mean of 1 and a standard deviation of -0.1. These noise levels were chosen empirically based on the observed data.

We applied the same prescreening process used in the previous study[30] to these synthetic events and conducted the Bayesian inversion for the accepted events. The results show that the prescreening pro-cess tends to reject events at some locations outside the station cov-erage, as indicated by open circles in Supplementary Fig. 8. Nevertheless, these rejected events were located far from the updip limit of the tremors identified in the current study. Therefore, we concluded that the updip limit of the tremors is robust. The synthetic test results also demonstrate that any tremor can be located with reasonable uncertainty (i.e., the 95% confidence interval <10 km) if it occurs outside the coverage area of region C.

## Multichannel seismic reflection profile

We use MCS data acquired from the KR02-11 cruise conducted in 2002 by the R/V Kairei of Japan Agency for Marine-Earth Science and Tech-nology. The data were collected using a ~5 km long hydrophone streamer cable with 25-m spaced receivers. An air-gun array with a total volume of 196.6 L (12,000 in³) was fired every 50 m along the survey lines. The towing depths of the streamer cable and the air-gun array were 20 and 10 m, respectively.

The reflection data were reprocessed using the latest standard processing techniques by DownUnder GeoSolutions (DUG) in Kuala Lumpur, Malaysia. Similar to preceding studies[64,65], a preprocessing sequence was applied to the data to reduce the noise and multiple reflections, including a tidal static correction, swell-noise attenuation,

de-ghosting, de-signature, surface-related multiples suppression, and parabolic Radon transform filtering. Velocity model building using reflection tomography and pre-stack Kirchhoff depth migration was performed to produce the final seismic reflection profiles.

In this study, we investigate seven lines shown in Fig. 3 and Sup-plementary Figs. 2–7. We track the key horizons of the top of the oceanic crust and the décollement as determined by previous studies[20,47,66] and interpret faults in the accretionary prism based on displacement or discontinuity of the seismic reflectors.

## Bathymetry data

To effectively extract structural information from the bathymetry of the study region, we utilized a publicly available de-trended bathy-metric digital elevation model (DEM) specific to our study area. This de-trended DEM is structured on a 52 m gridded space and is gener-ated by the filling of artificial sinks present in the original DEM and the subsequent application of a 15 km high-pass filter. Further details can be found in Schottenfels and Regalla's research article[35].

## Increase in the normal stress across the kink

We use a simplified two-dimensional model to illustrate how a geo-metrical barrier (i.e., an abrupt change in dip angle) affects slip pro-pagation. The model is defined in an $x$-$z$ coordinate system, with the $x$-axis aligned along the décollement (positive downward), and the $z$-axis oriented perpendicular to it (positive upward), as shown in Supplementary Fig. 11. With this coordinate, the unit normal vector to the décollement is given by $\mathbf{n} = (0, 1)^T$.

Assuming that the maximum principal stress axis is oriented at angle $\psi_b$ from the x-axis, the stress tensor is expressed as

$$\Sigma = \begin{pmatrix} \cos\psi_b & -\sin\psi_b \\ \sin\psi_b & \cos\psi_b \end{pmatrix} \begin{pmatrix} \sigma_1 & 0 \\ 0 & \sigma_3 \end{pmatrix} \begin{pmatrix} \cos\psi_b & \sin\psi_b \\ -\sin\psi_b & \cos\psi_b \end{pmatrix}$$

$$= \begin{pmatrix} \sigma_1\cos^2\psi_b + \sigma_3\sin^2\psi_b & (\sigma_1 - \sigma_3)\sin\psi_b\cos\psi_b \\ (\sigma_1 - \sigma_3)\sin\psi_b\cos\psi_b & \sigma_1\sin^2\psi_b + \sigma_3 \end{pmatrix}, \quad (1)$$

where $\sigma_1$ and $\sigma_3$ are the maximum and minimum principal stresses, respectively. A normal stress acting on the décollement is calculated as follows:

$$\sigma_n = \mathbf{n}^T\Sigma\mathbf{n} = \frac{\sigma_1 + \sigma_3}{2} - \frac{\sigma_1 - \sigma_3}{2}\cos 2\psi_b \quad (2)$$

This expression shows that $\sigma_n$ increases monotonically with $\psi_b$ for $0° < \psi_b < 90°$.

The stress orientation $\psi_b$ can be estimated using the Coulomb wedge theory[41]. Assuming a non-cohesive wedge, stress orientation is given by

$$\psi_b = \frac{1}{2}\arcsin\left(\frac{\sin\alpha'}{\sin\phi}\right) - \frac{1}{2}\alpha' + \alpha + \beta, \quad (3)$$

where the modified surface slope angle $\alpha\prime$ is defined as

$$\alpha' = \arctan\left[\left(\frac{1 - \frac{\rho_w}{\rho}}{1 - \lambda}\right)\tan\alpha\right] \quad (4)$$

Here, $\alpha$ is the surface (seafloor) slope angle; $\beta$ is the dip angle of the décollement; $\rho$ and $\rho_w$ are the densities of the wedge and seawater, respectively; $\lambda$ is the pore-fluid pressure ratio; and $\phi$ is the internal friction angle, defined as

$$\phi = \arctan(\mu_w), \quad (5)$$

where $\mu_w$ is the friction coefficient within the wedge.

Based on parameter values reported for the same region[42] (Supplementary Table 1), the resulting $\psi_b$ values are approximately 5° for the ITZ and 26° for the FTZ. From Eq. (2) and assuming that both the mean stress $(\sigma_1 + \sigma_3)$ and the differential stress $(\sigma_1 - \sigma_3)$ vary only gradually along dip, the normal stress at the FTZ exceeds that at the ITZ by about 20% of the differential stress. This partly supports the interpretation that the kink can act as a geometrical barrier to SSE propagation.

## Data availability

The raw MCS data can be requested from the JAMSTEC Seismic Survey Database[67] (https://doi.org/10.17596/0002069) using the cruise ID KR02-11. The tremor probability is available via Zenodo (https://doi.org/10.5281/zenodo.11646923). The bathymetry data used in Figs. 1 and 5 is available from the Japan Coast Guard website (https://www.jodc.go.jp/jodcweb/JDOSS/infoJEGG.html). The tremor and VLFE epicenters used in Supplementary Figs. 1, 9, and 10 are available from the Slow Earthquake Database[68] (http://www-solid.eps.s.u-tokyo.ac.jp/~sloweq/) and correspond to the Akuhara2023-Tremor, Hendriyana2021-Tremor, and Yamamoto2022-VLFE catalogs, respectively.

## Code availability

A computer program that has been used for locating tectonic tremors is available at https://doi.org/10.5281/zenodo.8333346.

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

## Acknowledgements

This study was supported by Cooperative Research Program of Atmosphere and Ocean Research Institute, The University of Tokyo (Research Vessel Shinsei-maru, JURCAOSS19-27 and JURCAOSS21-31); JSPS KAKENHI Grant Numbers 24H01021 and JP21H05202 in the Grant-in-Aid for Transformative Research Areas (A) "Science of Slow-to-Fast Earthquakes", awarded to T.A. and T.Ts., respectively; the Ministry of Education, Culture, Sports, Science and Technology (MEXT) of Japan, under its Third Earthquake and Volcano Hazards Observation and Research Program (Earthquake and Volcano Hazard Reduction Research).

## Author contributions

T.A. contributed the conceptualization of the study. T.A. and K.S. contributed to the analysis and visualization. T.A., K.S., T.Ts., A.H.F., R.Ar., T.To., and S.T. contributed drafting. T.A., Y.Y., H.S., A.H.F, and S.O. contributed to the data acquisition (passive-source data). R.Ar., G.F., and Y.N. contributed to the data acquisition (active-source data). T.A., Y.Y., H.S., Y.I., E.A., R.Az., R.H., K.M, S.T., T.Y., and M.S. contributed to planning the observations. All authors agreed with the final manuscript.

## Competing interests

The authors declare no competing interests.
