## [Transparent Peer Review file · Nature Communications]

Structural barriers control the spatial extent of slow earthquake slip

Corresponding Author: Dr Takeshi Akuhara

Version 0:

Reviewer comments:

Reviewer #1

(Remarks to the Author)

Review comments on "Geometrical barrier determines the updip limit of slow earthquake slip"

This manuscript provides the time-spacio relationships between precise shallow tremor locations and deformation structures from seismic profiles in the Nankai Trough off Kumamoto, SW Japan. Authors found that the frontal thrusts with higher dip angle are the barrier for the tremor, major place for the tremor is the horizontal decollement, tremor is inhibited in the seamount subduction area, and strike slip faults has an effect for the tremor migration. Authors show clearly that the shallow tremor distributions are strongly related to the deformation structures in subduction zone, which is very important to assess the scale of earthquake disasters for other subduction zones.

There are, however, some unclear descriptions of structures and some unclear interpretations in the discussion section. The relationships between structures, stress and strain (therefore the fluid pressure) are not so clearly mentioned in terms of their mechanisms. Because the probability of tremor distribution was already published in the previous study, the new things in this manuscript is the relationship of tremor locations to the deformation features in subduction zones. Therefore, the descriptions for the structures must be carefully conducted. The major purpose of this manuscript is to discuss about the mechanism how the tremor locality is controlled by the structure. But discussions do not so much get into the mechanisms but just follow the fact without any logical interpretations about the mechanisms in my impression. Especially, how the fluid pressure was controlled by the structures is not so clearly mentioned.

I will show a list of my comments below for the detail. My recommendation is major revision for this manuscript. As I mentioned above, the clear spatial relationships between shallow tremor and structures are very interesting and important. Therefore, I think this manuscript is worth to be published in Nature communications after the revisions are properly conducted.

L30: Pore fluid pressure contrast is not clear description. Authors need to describe it as that high fluid pressure is expected in the underplated structure.

L39: "occurrences" instead of "occurrence".

L91-92: It is not clear for me that the different trends of anticlines between ITZ and FTZ (Only in R3?). Please show them in Fig. 2. This difference in the trends is important in the discussion?

L114-115: Please show the reflections from oceanic crust and decollement in Fig. 3 to emphasize the one from oceanic crust and the other from decollement. The difference is very important. Authors want to say the difference the frontal thrusts between branching from the decollement (D1-D2) and not branching but the angle getting higher (D3-D5)?

L119: I agree that the reflectors branching from the decollement are the frontal thrusts which can not be active currently because the active decollement exists below. But the reflectors without decollement from which they are branching can be called decollement as a currently active plate boundary fault. Please define the frontal thrusts and decollement. Could be better to call the highly dipping thrust as a descriptive term.

L120: It is better to delete "may". It is clear to have decollement to the current deformation front in D1 and D2. But I am wondering that authors can say the proto-decollement is there in D3-D5. If there, you can say the reflectors with high dip

angle in D3-D5 as frontal thrusts.

L142: Branching points are there only in D1 and D2. Authors need to describe precisely these for all from D1 to D5.

L149: For the interpretations about increase in shear strength, explain it more detail. You need to say it with the regional stress state that has σ_1 is sub-horizontal. And the increase in shear strength itself does not mean the factor to prevent slip. Discussion for magnitude of differential stress is needed to evaluate enhancement or prevention of slip. Slip tendency is also another expression to evaluate slip enhancement or prevention. Rephrase the sentence to say how the slip is prevented with the higher angle of the thrusts.

L155: For the interpretations of less strain accumulation in the frontal area, why you can expect the less strain accumulation as a mechanism controlled by geometry? Please explain it. This is one of my concerns that authors say just a phenomenon without any mechanical interpretation.

L157: Delete "may".

L168-179: Please give a logical explanation with evidence for the interpretation.

L173: You do not need the interpretation for the LVZ as underplating. No evidence for underplating because duplex structure is not clearly observed in the structure and this interpretation can not be useful for your discussion. It is better just to say the transparent area has low velocity suggesting the high fluid pressure. It is enough.

L177: Why authors can say that the underplating occurs at the horizontal area? And such low velocity zone is not common every seismic profiles. If so, authors can not say that underplating structures correspond to the tremor distribution. As authors also pointed out below, the LVZ is not less visible in the eastern side. Really less visible but is it there? Show it.

L179-180: How to explain the fluid pressure controlled by the structures mechanically? Authors just describe the velocity distribution. No mechanical interpretation is provided for the relationship between structure and fluid pressure. I was confused between the horizontal decollement, underplating structure, high fluid pressure, frontal thrusting and less strain accumulation in the frontal area. What are the causes and the results? And how they are mechanically related?

L192: Why the stress shadow makes fluid pressure increased? Please explain it or cite previous studies.

L199: What does "locally overwhelmed the stress shadow condition" mean? Fluid pressure was increased in the stress shadow area locally by the intra slab earthquake? Why? Liquefaction?

(Remarks on code availability)

Reviewer #2

(Remarks to the Author)

The authors present high-resolution tremor locations for the Nankai Trough of Japan dating to 2020-2021, obtained using a dense networks of ocean-bottom seismometers and their own recently-developed technique for improved tremor detection and location. This region is extremely well characterized using geophysical and bathymetric imaging, among the best-known offshore subduction zones in the world. They interpret the boundaries of the tremor episodes in light of the well-delineated tectonic structure of the wedge sediments and underlying plate topography, including a prominent subducted ridge that has been imaged previously and has a major effect on the structural development and geometry of the accretionary wedge.

They show clearly and effectively that the tremor patch appears to be confined to the "ITZ," the portion of the outer wedge inboard of the frontal thrust zone. The updip limit of tremor in particular is spatially correlated with the branch point of a major frontal thrust splay fault system, and furthermore the along-strike propagation of the tremor episode appears to be limited or affected by the presence of a system of prominent strike-slip faults crossing the wedge. I find the argument to be clear and convincing. It is presented well, with an economy of data and figures. The result is important for Nankai structure and hazard specifically, but also for our global understanding of the nature of subduction zone megathrust strain accumulation and release, and the possible limits to co-seismic rupture.

There is essentially nowhere else in the world besides Nankai where tremor and SSE at the up-dip end of the locked system have been seen to anything approaching this extent and level of detail and resolution. I would enthusiastically support publication of this manuscript with only minor editorial revisions. Here are few comments for the authors and editors to consider:

The authors don't say much about the downdip limit of the tremor patch, but it seems to be correlated in map view with the seaward edge of the Kumano forearc basin, which has previously been identified as a potential mega-splay fault and up-dip limit of the main locked zone. Perhaps it would strengthen this manuscript to discuss the down-dip limit briefly as well as the up-dip limit? I think the case is compelling that the patch corresponds to the low-velocity, high-porosity zone identified in other papers that are cited here. To me, this is the most likely reason the tremor patch exists.

Throughout the manuscript: In the majority of papers on the region, the ridges are referred to as the “Zenu Ridge” and the “Paleo-Zenu Ridge.” I realize of course that zenu is a Japanese form of the foreign word zenith, so the usage in this manuscript is not incorrect. However, perhaps for clarity it would be useful to call it Paleo-Zenu, because many researchers are more familiar with that name?

Line 40: maybe not say “better” insights? Maybe “important” or “useful” insights would be more precise word choice.

Lines 158 – 164 and figure 4: There is a mention here of a 2016 episode of activity, in addition to the analyzed 2020-2021 sequence, but as far as I can see it is not mentioned elsewhere in the manuscript at all, so we don’t have much information on it, except two cited references. Perhaps another sentence or two on this sequence could be added, for better context and comparison? Otherwise, it could perhaps be omitted, since it doesn’t add to the point of this manuscript without context.

Lines 207-210: Papers by Sakaguchi et al. (2011) and Yamaguchi et al. (2011) analyzing core samples from IODP drilling of the Nankai frontal thrust both concluded that coseismic slip inducing frictional heating has affected very shallow regions in this zone (equivalent to along line D5, approximately). This is evidence that megathrust earthquake slip does propagate through the tremor zone, and should be mentioned here as well.

Figure 2. The black contours of tremor probability overlain on the rainbow bathymetry and heavy black fault traces in 2A are hard to clearly see and interpret. Perhaps a modified color scheme would be useful here. It’s not clear to me that the color representation of bathymetry in A conveys much information that is not also shown in B as a grayscale, so maybe just use that palette for both, with color contours of tremor for A? Also, in 2B, the blue solid and dashed lines are not identified in the caption.

Figure 4. The “slip-inhibitive” purple dashed line is described in the key as “on the megathrust” but is shown overlain on the frontal thrust, not the megathrust, except for in the panel labeled D7. This is a bit confusing, although I think that I understand what the authors intended, which is that slip transfer to the branching frontal thrust inhibits the tendency for slip on the underlying, lower-dip megathrust (décollement). I suggest modifying this figure and its key to make a bit more clear what is being highlighted.

Figure 5. The blue dashed hyperbola in panel C appeared at first look to be a set of blue dots, about the same size as the red and orange dots, at least in the provided pdf file. I could only see that it was actually a dashed line when I enlarged the figure to a very large size on screen. Perhaps the figure can be edited to more clearly distinguish the dashed line from the dots even at low magnification.

(Remarks on code availability)

Reviewer #3

(Remarks to the Author)

Review of Nature Communications ms. NCOMMS-24-45146-T – “Geometrical barrier determines the updip limit of slow earthquake slip” by Akuhara et al.

In this well-written manuscript the authors present high-resolution tremor locations, bathymetry, and seismic reflection images that place new constraints on the up-dip limit of shallow slow slip earthquakes in the Nankai subduction zone. In most of the study region, the up-dip limit of tremor is spatially correlated with the intersection of the subduction décollement and the main frontal thrust suggesting that changes in fault geometry and/or associated changes in fluid pressure may act as a barrier to slow slip. This finding has important implications for the up-dip limit of megathrust earthquakes and tsunami generation I recommend publication of this manuscript in Nature Communications after considering my minor comments on the manuscript and figures below.

- Labelling of regions R1-R3 and profiles D1-D7 in text and figures -- I suggest modifying the labelling of the regions and profiles to make it easier for the reader to understand which profiles correspond to which regions. Based on my understanding, profiles D1 and D2 occur in region R1; profiles D3, D4, and D5 occur in region R2; and profiles D6 and D7 occur in region R3. As one possible approach, if the regions were labelled A, B, and C, then the seven profiles could be labelled A1, A2, B1, B2, B3, C1, and C2 making it easy to see which profiles correspond to which regions.

Line 95 – I suggest changing “crosscut” to “intersect”. From the information presented in this paper, it is not clear to me that the right-lateral strike-slip faults offset the décollement or are restricted to the upper plate.

Line 135 “Influence of geometry and pore fluid pressure variations on slip propagation” section -- In this section, I recommend adding a few sentences comparing the new results from Nankai presented in this manuscript with what has been discovered about the up-dip limit of shallow slow-slip earthquakes in other subduction zones, such as the Hikurangi, New Zealand (e.g., Wallace et al., 2016, Science) and Costa Rica (e.g., Davis et al., 2015, EPSL) subduction zones.

Lines 147-149 – I was not able to follow why the change in dip angle would increase the normal stress. I can understand how the change in geometry would result in a geometric barrier to slip and arguably the shear stress that would need to be overcome would increase from the décollement to the frontal thrust. But with the steeper dip (and shallower depth) of the

frontal thrust I would expect the normal stress on the faults to decrease toward the trench. I recommend the authors' clarify this point to the extent possible

Paragraph starting at line 171 – I suggest adding a reference to Wallace et al. (2012, Geology) or similar paper that has proposed variations in fluid pressure as affecting the slip behavior of subduction plate interfaces. If high fluid pressures along the decollement are required for slow slip to occur, then it seems plausible that fluid could be leaking up the frontal thrust, reducing fluid pressures at the intersection with the decollement and therefore providing an up dip limit to slow slip.

Line 184 – I suggest changing “overwrite” to “cut across” or “obscure”.

Line 211 – I agree with the authors that the geometrical barrier () identified in their study might prevent a future megathrust earthquake from reaching the trench, but it is also possible megathrust displacement along the decollement may track up the frontal thrust and still displace the seafloor.

Figure 2 – The tremor probability contours in Figure 2a cannot be deciphered at the scale of the figure. If this figure cannot be expanded in the main paper, I recommend that only 1-2 key contours be shown and the expanded figure with more contours be included in the supplemental material.

Figure 2 and 5 – I recommend adding horizontal scales to these figures.

(Remarks on code availability)

Version 1:

Reviewer comments:

Reviewer #3

(Remarks to the Author)

All of my concerns with the original manuscript have been appropriately addressed by the authors in the revised manuscript and I recommend publication in Nature Communications.

(Remarks on code availability)

Reviewer #4

(Remarks to the Author)

Dear Editor,

As was pointed by previous reviewers, the manuscript by Akuhara et al. presents an interesting correlation between a slow slip-tremors event in Nankai Trough and the geometrical features of the interplate boundary and continental wedge structures observed in high quality active seismic lines. This observation is important for the open research line associated to the interplay between seismotectonics and structural/geodynamical/geological evolution of subduction zones. As I'm participating in a second round of revision, I give some comments of the last version of the manuscript, considering only the points that, in my view, were not previously solved in the revision process. Below, I give comments after an R, referencing the corresponding lines in the text and the beginning of specific paragraphs.

Lines 75-76: “For comparison, the previous tremor episode...” R: Please include the location of this previous tremor episodes in a map figure. This should be important for interpretation.

Lines 92-93: “... outer wedge situated on the trenchward side of a megasplay fault...” R: In Fig.2 it is clear that the correlation between the megasplay fault and the updip of tremor events is even better than the correlation between the updip of this activity and the “kink” barrier. In my opinion profiles figures and text should be modified to include a more complete discussion of the updip relation between tremors and continental wedge structure.

Lines 93-95: “... The outer wedge is divided into two geological 93 zones...” R: Please include the limit between ITZ and FTZ in figure 2. This should help to note that the updip limit of slow earthquake is correlated with this limit only in the zone B.

Lines 96-97: “...two subparallel right-lateral strike-slip faults located in the ITZ...” R: Where are this right-lateral structures?. Please include the reference to the corresponding Figures.

Lines 114-119: “...to region C, which lacks such features, aligns with the eastern boundary of the tremor distribution.

Furthermore...” R: C is the eastern limit of this tremor episode, but this does not mean that future tremors episodes could not be located seaward in the segment C. To interpret that, the authors should analyze a much longer period of seismicity along the margin. In my view the analysis do not prove the tectonic control of the updip seismicity in C.

Lines 154-155: “...current study indicate that the propagation of SSEs along the décollement is hindered at the kink or branching points...” R: Respecting to figure, the correlation between the updip of SSEs and the kink or branching points is clearly observed at the profiles B and in the profile A2. However, analyzing the profile A1 (in the supplementary material) the

“kink” seems to be located landward from the point interpreted by the authors, then at the profile A1 the correlation with the updip of SSEs is not clear. Geometrically, the “kink” points seem to be correlated with the landward limit of the FTZ high. Maybe this observation is debatable, but in any case, it is important the inclusion of figures of the seismic profiles without interpretation (in the supplementary material) to facilitate the observation of reflectors near the decollement.

Lines 159-164: “...the steep thrust experiences higher normal stress than the gently dipping section of the décollement...” R: Respecting the discussion of Normal stresses, the authors include an analysis that is correct under the assumptions of the horizontality of the maximum stress and constant frictional conditions. However, both assumptions are difficult to associate with an accretionary wedge deformed under compressive forces, gravitational forces and different internal and basal effective friction coefficients. In fact, the formation of a kink should be related precisely with a more complex stress field inside the wedge (see for instance the simple model of Dahlen et al. 1984, Noncohesive critical Coulomb wedges: An exact solution). On the other hand, the observation of the current and paleo deformation fronts should be related with variations of the basal effective friction coefficient and maybe with the aseismic deformation of sedimentary units near the deformation front. In my opinion, a more detailed discussion of the stress and strain field in accretionary wedges should be considered for the interpretation of structures and the SSEs episode.

Lines 237-239: “...This lack of deformation is likely due to a stress shadow caused by the subduction of the Paleo–Zenisu ridge...” R: where is the Palo Zenisu_Ridge? Please indicate in maps and profiles. What is the definition of stress shadow used here? The authors of the reference 47 defined shadow zones seaward from subducted oceanic features and below inactive decollements, which seems to be not the case of profiles C. What is the subducted oceanic feature (observed in the profile) that generates the “stress shadow”? Please indicate in maps and figures.

Supplementary material: There are some signs “Error!” in figure captions.

(Remarks on code availability)

Reviewer #5

(Remarks to the Author)

Review by Madison Bombardier of the manuscript “Geometrical barrier determines the updip limit of slow earthquake slip” by Akuhara et al.

This manuscript uses a previously-published tremor catalogue to examine the relationship of shallow slow slip (using tremor as a proxy) to subduction zone structures imaged by seismic reflection in Nankai. Their main finding is the interpretation that the updip limit of slow slip is bounded by an abrupt steepening of the main thrust fault (top of decollement) in most of the region. They infer that the impedance of slow slip on the steep frontal thrust segment may imply impedance of coseismic slip during regular earthquakes, which may have consequences on tsunami generation and associated hazards.

Three reviewers during the first round of peer review requested: (1), clearer and more robust mechanistic explanations of slip impedance along the steep frontal thrust in terms of pore-fluid pressure and stress; (2), discussion of the downdip limits of tremor; and (3), many editorial changes. The authors have reasonably addressed most of the comments from these reviewers, with some exceptions (see my comment number 29 below); however, some explanations and claims are vague or poorly supported.

The work presented in this manuscript is important to the understanding of subduction zone science, and represents an effective use of tremor catalogues for the study of this topic. After minor revision considering my comments below, this manuscript is worthy to be published. I have two main comments, and more than 3 dozen minor comments.

(1) Main comment:

The authors claim in multiple locations throughout the manuscript that the tremor catalogue they use contains “precise” tremor locations; however, evidence for this claim is not provided. The only relevant information provided is that the horizontal uncertainty on each tremor epicentre is estimated by one half the width of the 95% confidence interval to be “a few kilometres” (lines 72–73). It is not clear to me that this is precise, or even more precise than other methods, nor do I think that such a claim is necessary. The primary advantage of Bayesian inference is that solutions are provided as probability distributions, which gives unique 3D probability distributions of each tremor event from which uncertainty can be quantified; there is no guarantee of precision. In other words, this catalogue provides robust uncertainty estimates. Because of this, the tremor catalogue may be considered reliable or credible, but not precise. This is especially true in the context of this analysis where multi-channel seismic reflection data is used with resolution on the order of 100 m (line 82).

Despite that the primary strength of this tremor catalogue is the detailed uncertainty estimation, the authors do not provide quantitative estimates or distributions of these uncertainties, nor do they consider them in their analysis. Supplementary figure 1 shows all tremor epicentres with 95% confidence interval error bars, but the superposition of the densely-clustered events obscures most of this information.

Here I make several suggestions to address these issues:

- remove descriptors such as “precise” everywhere related to the tremor catalogue
- in the main text, replace “a few kilometres” (lines 72–73) with mean or median horizontal uncertainty estimates, and/or state the range within which 50% or 75% or 90% (or whatever) of horizontal uncertainties are in.
- at the location where the tremor catalogue is introduced in the main text, briefly explain the advantage that Bayesian inference provides to this catalogue in terms of robust uncertainty estimation. Probabilistic solutions are used to represent the entire catalogue as one probability map, which provides more information than a scatterplot of point sources. A probability map is an unusual way to represent a catalogue of seismic events, and you should clearly explain the importance and novelty of this to encourage readers to adopt similar methods in future work.
- change supplementary figure 1 to better represent the geographic distribution of uncertainties. I suggest to segment the area into 1 km (or 5 km, or whatever) squared cells and colour each cell according to the average (or median) uncertainty of tremor events in that cell. In this case, you would need two panels in the figure: one for NS uncertainty and one for EW uncertainty. This method would allow the reader to view the heterogeneous geographic distribution of uncertainty over the study area. I expect average uncertainties vary systematically over the region, which may be useful information for the main analysis. In particular, I wonder about the EW uncertainties on the downdip tremor patch in the far north of the study area (region C). This type of information should be clearly discernible from this figure. An example can be found in the supplementary information of Bombardier et al 2025 (doi: 10.1016/j.tecto.2025.230752).

(2) Main comment:

Interpretations made in this analysis require a framework of SSEs wherein slow slip directly generates/triggers tremor, and wherein both tremor and slow slip occur on the megathrust fault (in this manuscript, the top of decollement). It is not my understanding that this framework is universally accepted as certain fact, thus the high degree of certainty with which it is presented is not warranted.

Regarding lines 60–62: it has been shown that tremor does indeed tend to occur near the slip front in Cascadia (eg Hall et al 2019, Bartlow et al 2011, Luo and Liu 2019), but I do not think there is sufficient evidence to use the word “triggered”, since there may be multiple processes occurring at the slip front that may cause, generate, or trigger tremor. In addition, shallow slow slip may not be perfectly analogous to deep slow slip where this has been studied more extensively. The brittle-asperity-in-ductile-matrix model (Luo and Liu 2019) is a plausible hypothesis (among others) to explain some elements of tremor and slow slip, and using this framework is justifiable, but it should be presented as a framework or base-level assumption rather than a known fact about the physical mechanisms.

Regarding lines 149–153: this reasoning is fuzzy. If you assume tremor is directly caused by slow slip, as the previous sentence states, how can the depth of tremor “most likely” be “similar” to that of SSEs? Would it not be certain that the depths are exactly the same? Reference 19 (Toh 2018) states that some VLFEs likely occurred on a splay fault, which indicates that the depths of these VLFEs are not localized entirely to the megathrust fault (insofar as depth constraints on VLFEs support such an observation); or that the location/structure of megathrust fault itself is uncertain or complex. In addition, deep tremor in Cascadia represented as probability resulting from a Bayesian location method (Bombardier et al 2025) indicates that the depth of tremor can deviate significantly from the top of the subducting oceanic crust and is inconsistent along the margin. Again, deep tremor/SSEs may not be so analogous to shallow tremor/SSEs, but given the difficulty of adequately constraining depths of tremor/LFEs/VLFEs/slow slip, and the uncertainty regarding the relationship between these phenomena, consideration that these processes may occur beyond the megathrust fault is warranted. The possibility of tremor and SSE activity beyond the megathrust fault has important implications for discerning active structures and understanding hazards, and the study of this topic should not be discouraged by premature or simplifying assertions.

My suggestion to address this issue is to clearly explain that the analysis and interpretations presented in this manuscript requires the assumption of a specific framework/hypothesis about the nature of the relationship between tremor and slow slip, and about which structures host this activity due to the fact that tremor depths are not resolved in this catalogue. You already have many references to use to justify the use of such a framework. Also explain that variation exist between regions globally and uncertainty regarding the depth and host structures for such activity is prevalent, more work on these topics is needed. These changes largely involve reframing and rephrasing at various locations in the manuscript.

(3) The abstract should include why your results matter. I suggest simplifying the summary of results in the abstract to make space for the broader interpretations/impact of the results mentioned in the “introduction” and “impacts” sections.

(4) Lines 51–53 it may be relevant to the present manuscript that the forearc mantle corner has also been shown as a boundary to slow slip by Sawaki et al (2021) and Bombardier et al (2024).

(5) Lines 99–100 regarding a new deformation front, this should have a citation.

(6) Line 101 the tremor probability map is mentioned for the first time without explanation of what it is. A reference to the methods section where the probability map is mathematically defined is missing.

(7) The ITZ-FTZ boundary denoted in Figure 2b by the purple dashed line would be helpful in panel (a) of this figure, so that the spatial correspondence between this boundary and the updip limit of tremor is very clear.

(8) It is not that easy to discern the annotations in Figure 2b. The dark colour of the bathymetry shading makes it difficult to clearly see the thin black font of the text and transparent blue/red lines. It would be better to make these annotations more pronounced, possibly by decreasing their transparency, making the text font bolder, and/or increasing the transparency of the grey bathymetry shading.

(9) Lines 115–117 here is an example where it is important to know the uncertainties on tremor in region C. “much further landward” may not be a well-supported observation depending on the uncertainties.

(10) Lines 118–121 This sentence is long and awkward, and should be in present tense rather than in a past tense narrative-style description of what you did. For example, try “To explore the apparent agreement between the tremor distribution and bathymetric features, we [analyze]...”

(11) Supplementary figures 2–4 have a “Error! Reference not found” note in the captions. Also the caption in supplementary figure 2 should reference “inverted triangles” (plural), not “inverted triangle”. These inverted triangles in profile A1 are difficult to see clearly; I did not realize they were triangles at first. Perhaps make them bigger with a black outline?

(12) Lines 124–125 “almost subparallel” doesn’t really make sense. I suggest rephrasing as “almost parallel” or simply “subparallel”. In addition, this short sentence could be joined with the previous to make it less awkward. For example, “The most prominent features ... oceanic crust and decollement, which are subparallel within the ITZ.”

(13) Line 131 I suggest changing “new” and “old” deformation front to “deformation front” and “mega splay”. To say “new” implies it is a new interpretation in this manuscript, and it can confuse the reader when there is no discussion of this new feature. This requires changes to annotations in figures as well.

(14) Lines 143–144 Supplementary figure 4 does not actually show the tremor distribution along profile C1 and C2. The map in Figure 3 is a more appropriate reference for this observation. However, as a reader, I find that I need to refer to both figure 3 and supplementary figure 4 to fully understand this observation, which is cumbersome. It would help to include the tremor probability distribution along each profile in supplementary figures 2–4 (as is done in figure 3).

(15) I suggest including an annotation in supplementary figure 4 to identify the subducting ridge. I can guess what constitutes the ridge, but better to identify it clearly because I may be wrong.

(16) Lines 153–155 this is the only location figure 4 is referenced. Perhaps this early reference to this figure should be removed so the figures can be ordered logically; I do not think it adds much at this point in the manuscript. Figure 4 represents a summary of your interpretations, so it should be explained near the end of the manuscript and used to support such a summary. Notably, a brief summary of the new observations and interpretations do not really exist, but are apt.

I think lines 153–155 should say “... the propagation of SSEs along dip is hindered...” since figure 4 also shows the ridge stress shadow and the strike slip faults hinder along-strike SSE propagation.

(17) Lines 155–158 reference 40 (Wech and Bartlow 2014) is a study from Cascadia that observed one instance of an SSE occurring without tremor for a short time. While I agree that these cases seem infrequent in Cascadia, I do not think this is relevant to shallow SSEs in Nankai. It is reasonable to exclude SSEs without tremor, but this should be explained and justified near line 60 where the framework, assumptions, and unknowns are explained. And this choice should be supported with observations for the relevant region. Also, related to my main comment (2), SSEs occurring without tremor are incompatible within a framework wherein slip generates tremor, as stated just three sentences before this. If there are SSEs that occur without tremor, such that they are excluded from this analysis, maybe the framework needs to be revised?

(18) Line 175 “distinctly different” is a redundant phrase. “Distinct” and “different” are synonyms.

(19) Line 176 rephrase this as “A stratified pattern observed at the trench side of the steep thrust is absent on the landward side.” and it would be better to reference figure 3 rather than supplementary figure 2, since this feature can be seen in the main-text figure.

(20) Line 185 “in distance” is unnecessary because kilometre is a unit of distance.

(21) Line 186 briefly explain why the slip is diffusive, and why diffusive slip is characterized by a hyperbola.

(22) Line 191 “bounded” is different from “impeded”. In this manuscript, the tremor migration is impeded, not bounded by the strike slip faults.

(23) Line 197 “horizontally” is unnecessary.

(24) Lines 197–198 could slip have propagated along one of the strike slip faults? This could explain why the second small SSE initiated landward of the termination of the larger SSE (roughly along the strike of the strike slip faults).

(25) grey dots in figure 5b are not explained anywhere.

(26) Line 203 rephrase to “Although its spatial distribution is not well constrained, the potential role of pore-fluid pressure in the study region is worth discussing here.”

(27) Lines 205–206 the transparent facies are observed immediately above the decollement, not oceanic crust, according to figure 3.

(28) Line 211 rephrase to "... high pore-fluid pressure is interpreted to be sustained by ..."

(29) Lines 212–216 this section was added in response to comment 1-15 from the first review. It is awkward and oddly written as a string of short facts that are not presented cohesively. How exactly are these inferences relevant to the analysis? Explain "tectonic push" and what "these structures" are that may create tectonic push; does this refer to basement highs? Not clear.

(30) Lines 237–238 it is my understanding that subducting ridges and seamounts lead to highly fractured/deformed accretionary prisms/forearcs. In order for a lack of deformation in the prism to result from the stress shadow of a subducted ridge, it seems to me that the undeformed prism material would have to have been accreted after the ridge (in its shadow) in order to remain relatively undeformed. If the prism material was accreted first, then the ridge would have subducted under it, and the explanation given in the manuscript does not make sense to me. In this case, lack of deformation within the prism would not be a reliable indicator of the existence of a stress shadow. Please clarify the accretionary history and the relationship between subducted ridges/seamounts and prism deformation. Maybe there is an alternative explanation for the reflective strata?

(31) Line 252 rephrase to "... large earthquake may have facilitated tremor ..."

(32) Line 257 I believe the boundary between the inner and outer wedges is the megasplay fault shown in figure 2b and mentioned on line 93. I suggest including the "megasplay" term here with reference to figure 2b so it is clear.

(33) Line 258 "distinctly differ" is redundant

(34) Lines 262–263 please explain how a discontinuous stress state in the outer forearc wedge would affect slip on the megathrust.

(35) Line 264 the use of the term "separate" is confusing here. The tremor episode that is the focus of this manuscript occurs in the transient slip zone, so how can it separate the transient slip zone from the fully locked zone? I think this should be phrased "the downdip limit of this tremor episode may not be a universal representation, as SSEs and tremors have occurred further downdip in earlier episodes."

(36) Line 283 "numerous numerical" is a slightly awkward phrase. I suggest "multiple", "several", or any other synonym for "numerous".

(37) Line 294 I do not think Hikurangi is a relevant example here because tremor is not closely associated in space or time with shallow SSEs (eg Todd et al, 2018). As such, the tremor location methods used in this paper would be useful for locating tremor in Hikurangi, but not be useful for approximating slow slip.

(38) There are numerous grammatical errors and typos throughout the main text and supplementary material, in addition to the ones mentioned above. Text added in response to the earlier reviewers could be integrated better so that the text flows well and seems cohesive.

(39) Inconsistent use of "Figure" and "Fig" (eg line 212), and bolded text for figure references (eg line 206).

(Remarks on code availability)

The repository appears to contain adequate information for installation and use of the tremor location code. I did not attempt to install or run the code. Installation should be trivial since it is accessed through GitHub. I did not attempt to run the code because it requires a Fortran compiler and I have not had success with Fortran on modern MacOS.

Version 2:

Reviewer comments:

Reviewer #4

(Remarks to the Author)

Dear Editor,

All my concerns expressed in the previous review round were respectfully considered in the new version of the manuscript. In my view, the paper is ready to be published.

Regards

(Remarks on code availability)

Reviewer #5

(Remarks to the Author)

I previously reviewed this manuscript as reviewer # 5 and have been asked to confirm whether my comments have been addressed satisfactorily by the authors. I have read through the responses to my comments and the revised manuscript. I am happy with the changes that have been made, and I thank the authors for carefully addressing my many comments. I think the newest version is much stronger and cohesive than previous versions. The results and discussions presented are scientifically sound and important to the study of subduction zones. In particular, this work presents an effective application of tremor localization for the study of physical and structural properties of a subduction zone. I think this manuscript is ready to proceed to copy editing.

(Remarks on code availability)

Reply to reviewers' comments on the manuscript

“Geometrical barrier determines the updip limit of slow earthquake slip (NCOMMS-24-45146-T)”

Response to Reviewer #1:

Comment 1-1: This manuscript provides the time-spacio relationships between precise shallow tremor locations and deformation structures from seismic profiles in the Nankai Trough off Kumamoto, SW Japan. Authors found that the frontal thrusts with higher dip angle are the barrier for the tremor, major place for the tremor is the horizontal decollement, tremor is inhibited in the seamount subduction area, and strike slip faults has an effect for the tremor migration. Authors show clearly that the shallow tremor distributions are strongly related to the deformation structures in subduction zone, which is very important to assess the scale of earthquake disasters for other subduction zones.

There are, however, some unclear descriptions of structures and some unclear interpretations in the discussion section. The relationships between structures, stress and strain (therefore the fluid pressure) are not so clearly mentioned in terms of their mechanisms. Because the probability of tremor distribution was already published in the previous study, the new things in this manuscript is the relationship of tremor locations to the deformation features in subduction zones. Therefore, the descriptions for the structures must be carefully conducted. The major purpose of this manuscript is to discuss about the mechanism how the tremor locality is controlled by the structure. But discussions do not so much get into the mechanisms but just follow the fact without any logical interpretations about the mechanisms in my impression. Especially, how the fluid pressure was controlled by the structures is not so clearly mentioned.

I will show a list of my comments below for the detail. My recommendation is major revision for this manuscript. As I mentioned above, the clear spatial relationships between shallow tremor and structures are very interesting and important. Therefore, I think this manuscript is worth to be published in Nature communications after the revisions are properly conducted.

Reply 1-1: We would like to express our gratitude to the reviewer for the insightful comments. As the reviewer pointed out, the previous manuscript lacked detailed

information about the mechanisms of how slip propagation, structure, and stress interact with each other. In the revised manuscript, we have attempted to describe this as much as possible. Please refer to our point-by-point responses below.

Comment 1-2: L30: Pore fluid pressure contrast is not clear description. Authors need to describe it as that high fluid pressure is expected in the underplated structure.

Reply 1-2: We have rephrased the sentence for clarity (Lines 32–33 in the revised manuscript).

Comment 1-3: L39: “occurrences” instead of “occurrence”.

Reply 1-3: We have changed it to “occurrences” (Line 41).

Comment 1-4: L91-92: It is not clear for me that the different trends of anticlines between ITZ and FTZ (Only in R3?). Please show them in Fig. 2. This difference in the trends is important in the discussion?

Reply 1-4: The change in the trend is only seen in region C (previously referred to as R3), as the reviewer pointed out. Since this sentence is not necessary for our conclusion, we have removed this sentence.

Comment 1-5: L114-115: Please show the reflections from oceanic crust and decollement in Fig. 3 to emphasize the one from oceanic crust and the other from decollement. The difference is very important. Authors want to say the difference the frontal thrusts between branching from the decollement (D1-D2) and not branching but the angle getting higher (D3-D5)?

Reply 1-5: We consider that the reflections from the oceanic crust and décollement are already clearly indicated in the original Figure 3 using thick solid and dashed lines, respectively. The original sentence merely introduces to readers the primary features seen in the profile, that is, reflections from the top of oceanic crust and the décollement. To avoid potential confusion, we have rephrased the sentence for clarity (Lines 123–125).

Comment 1-6: L119: I agree that the reflectors branching from the decollement are the frontal thrusts which can not be active currently because the active decollement exists below. But the reflectors without decollement from which they are branching can be called decollment as a currently active plate boundary fault. Please define the frontal thrusts and decollement. Could be better to call the highly dipping thrust as a descriptive term.

Reply 1-6: We are grateful for this suggestion, and we have reorganized the usage of some geological terms. In the revised manuscript, the term “décollement” is used only for the active plate boundary. Consequently, what was previously referred to as “frontal thrust” along lines B1-B3, C1, C2 (formerly D3-D7) is now regarded as part of the “décollement”. The term “steep thrust” is used to denote the steeply dipping portion of the decollement (Lines 127–133).

Comment 1-7: L120: It is better to delete “may”. It is clear to have decollement to the current deformation front in D1 and D2. But I am wondering that authors can say the proto-decollement is there in D3-D5. If there, you can say the reflectors with high dip angle in D3-D5 as frontal thrusts.

Reply 1-7: We have rephrased the sentence. Since the proto-decollement is not clear in the image, we have decided to use the term “steep thrust” for lines B1-B3 (Lines 127–133).

Comment 1-8: L142: Branching points are there only in D1 and D2. Authors need to describe precisely these for all from D1 to D5.

Reply 1-8: Now the revised manuscript uses “kink” points instead of “branching” points for the eastern profiles B1-3 and C1-2. For profiles A1-2, we use “branching” points. The concerning sentence has been modified accordingly (Lines 153–155).

Comment 1-9: L149: For the interpretations about increase in shear strength, explain it

more detail. You need to say it with the regional stress state that has σ_1 is sub-horizontal. And the increase in shear strength itself does not mean the factor to prevent slip. Discussion for magnitude of differential stress is needed to evaluate enhancement or prevention of slip. Slip tendency is also another expression to evaluate slip enhancement or prevention. Rephrase the sentence to say how the slip is prevented with the higher angle of the thrusts.

Reply 1-9: We have renewed the corresponding paragraph (Lines 159–171) and explicitly stated that the maximum principal stress is sub-horizontal. Moreover, new paragraphs have been added to the Methods section (Lines 423–446) to provide a brief mathematical model explaining how the steep thrust can act as a geometrical barrier.

Regarding the use of slip tendency (or static differential stress), while it is a useful measure for assessing the likelihood of fault slip under static conditions, it may not be directly applicable to discussions of rupture termination, where dynamic shear stress plays a crucial role. Instead, we focus on shear strength following Bletery et al. (2016).

For readers interested in the slip tendency, we retain the sentence mentioning a previous study that investigated the slip tendency of this region (Hashimoto et al. 2022) (Lines 166–171). Overall, their results are consistent with our interpretation.

Comment 1-10: L155: For the interpretations of less strain accumulation in the frontal area, why you can expect the less strain accumulation as a mechanism controlled by geometry? Please explain it. This is one of my concerns that authors say just a phenomenon without any mechanical interpretation.

Reply 1-10: In the previous manuscript, we regrettably did not dedicate sufficient discussion to this interpretation. By referring to 'less strain accumulation,' the intention was to suggest the possibility that the décollement undergoes stable sliding. The well-stratified pattern observed in the hanging wall (Supplementary Fig. 2) appears to be consistent with stable sliding. We have incorporated this information into the main text (Lines 172–182) while ensuring that the interpretation remains non-conclusive.

Comment 1-11: L157: Delete “may”.

Reply 1-11: We have deleted it (Line 183).

Comment 1-12: L168-179: Please give a logical explanation with evidence for the interpretation.

Reply 1-12: The separation distance between the termination of the first migration and the initiation of the second migration is approximately 5 km (Line 196). This separation distance is relatively small in comparison to the length of the first migration (~40 km; Line 185), thereby indicating the potential for stress transfer from the first migration to trigger the subsequent migration (Lines 197–201). To enhance clarity, the relevant paragraph has been revised (Lines 192–201).

Comment 1-13: L173: You do not need the interpretation for the LVZ as underplating. No evidence for underplating because duplex structure is not clearly observed in the structure and this interpretation can not be useful for your discussion. It is better just to say the transparent area has low velocity suggesting the high fluid pressure. It is enough.

Reply 1-13: In accordance with the reviewer's suggestion, we have decided not to use the word "underplating" for this structure, acknowledging that the geological interpretation of this structure is still debated. We instead refer to it as LVZ in the revised manuscript.

Comment 1-14: L177: Why authors can say that the underplating occurs at the horizontal area? And such low velocity zone is not common every seismic profiles. If so, authors can not say that underplating structures correspond to the tremor distribution. As authors also pointed out below, the LVZ is not less visible in the eastern side. Really less visible but is it there? Show it.

Reply 1-14: We have substantially reorganized the paragraph concerning the spatial extent of the LVZ (Lines 217–230). We consider that the one-to-one spatial correspondence between tremor and high pore-fluid pressure (or LVZ) remains challenging, as the fluid pressure distribution is not well constrained even with the comprehensive set of reflection profiles. Nevertheless, the discussion of the potential role

of fluid pressure would be still valuable for readers because fluid pressure is recognized to regulate the slip behaviors of faults. This position is explicitly stated in the first line of the renewed paragraph.

With regard to the updip limit of the LVZ, we have removed the speculative sentence “underplating primarily occurs in areas where the thrust is horizontal”. Instead, we mention a possibility of the absence of LVZ above the steep thrust, by referring to some geophysical studies (Akuhara et al. 2020; Park et al. 2010; Azevedo et al. 2018; Lines 220–221).

For the LVZ in region A, we have included a detailed description of a previous study (Fahrudin et al. 2022; Lines 224–227), which interprets a negative reflector above the decollement as the top of the LVZ (or what they call “shear zone”). We have clarified this reflector in Supplementary Fig. 2. It is acknowledged that the clarity of this reflector is inadequate on the reflection image used in this study, and thus we avoid making conclusive statements in this paragraph.

Comment 1-15: L179-180: How to explain the fluid pressure controlled by the structures mechanically? Authors just describe the velocity distribution. No mechanical interpretation is provided for the relationship between structure and fluid pressure. I was confused between the horizontal decollement, underplating structure, high fluid pressure, frontal thrusting and less strain accumulation in the frontal area. What are the causes and the results? And how they are mechanically related?

Reply 1-15: We have included information about the relationship between fluid pressure and structure. Added information includes fluid source (Lines 210–211); how high pore fluid pressure is sustained (Lines 211–212); effects of basement high (Lines 212–213); and evidence showing that high pore fluid pressure is linked to the slow earthquake generations (Lines 213–216).

Comment 1-16: L192: Why the stress shadow makes fluid pressure increased? Please explain it or cite previous studies.

Reply 1-16: Throughout the manuscript, we interpret that the stress shadow “decreases”

the fluid pressure. We guess that the “increased” in the reviewer’s comment is merely a typo of “decreased” and therefore have added a detailed description how the stress shadow decreases fluid pressure compared to other regions (Lines 241–246).

Comment 1-17: L199: What does “locally overwhelmed the stress shadow condition” mean? Fluid pressure was increased in the stress shadow area locally by the intra slab earthquake? Why? Liquefaction?

Reply 1-17: The sentence was merely intended to convey that slow earthquakes were triggered by stress perturbation from the M7.4 earthquake. There is no requirement for fluid pressure. We have rephrased the sentence for clarity (Lines 251–253).

Response to Reviewer #2:

Comment 2-1: The authors present high-resolution tremor locations for the Nankai Trough of Japan dating to 2020-2021, obtained using a dense networks of ocean-bottom seismometers and their own recently-developed technique for improved tremor detection and location. This region is extremely well characterized using geophysical and bathymetric imaging, among the best-known offshore subduction zones in the world. They interpret the boundaries of the tremor episodes in light of the well-delineated tectonic structure of the wedge sediments and underlying plate topography, including a prominent subducted ridge that has been imaged previously and has a major effect on the structural development and geometry of the accretionary wedge.

They show clearly and effectively that the tremor patch appears to be confined to the “ITZ,” the portion of the outer wedge inboard of the frontal thrust zone. The updip limit of tremor in particular is spatially correlated with the branch point of a major frontal thrust splay fault system, and furthermore the along-strike propagation of the tremor episode appears to be limited or affected by the presence of a system of prominent strike-slip faults crossing the wedge. I find the argument to be clear and convincing. It is presented well, with an economy of data and figures. The result is important for Nankai structure and hazard specifically, but also for our global understanding of the nature of subduction zone megathrust strain accumulation and release, and the possible limits to co-seismic rupture.

There is essentially nowhere else in the world besides Nankai where tremor and SSE at the up-dip end of the locked system have been seen to anything approaching this extent and level of detail and resolution. I would enthusiastically support publication of this manuscript with only minor editorial revisions. Here are few comments for the authors and editors to consider:

The authors don't say much about the downdip limit of the tremor patch, but it seems to be correlated in map view with the seaward edge of the Kumano forearc basin, which has previously been identified as a potential mega-splay fault and up-dip limit of the main locked zone. Perhaps it would strengthen this manuscript to discuss the down-dip limit briefly as well as the up-dip limit? I think the case is compelling that the patch corresponds to the low-velocity, high-porosity zone identified in other papers that are cited here. To me, this is the most likely reason the tremor patch exists.

Reply 2-1: We thank the reviewer for the positive feedback. We agree with the importance of discussing the downdip limit. Thus, we have added a paragraph that briefly discusses the downdip limit (Lines 255–266).

Comment 2-2: Throughout the manuscript: In the majority of papers on the region, the ridges are referred to as the “Zenu Ridge” and the “Paleo-Zenu Ridge.” I realize of course that zenu is a Japanese form of the foreign word zenith, so the usage in this manuscript is not incorrect. However, perhaps for clarity it would be useful to call it Paleo-Zenu, because many researchers are more familiar with that name?

Reply 2-2: We have changed “Zenith” to “Zenu”.

Comment 2-3: Line 40: maybe not say “better” insights? Maybe “important” or “useful” insights would be more precise word choice.

Reply 2-3: We have changed it to “important” (Line 42).

Comment 2-4: Lines 158 – 164 and figure 4: There is a mention here of a 2016 episode of activity, in addition to the analyzed 2020-2021 sequence, but as far as I can see it is not

mentioned elsewhere in the manuscript at all, so we don't have much information on it, except two cited references. Perhaps another sentence or two on this sequence could be added, for better context and comparison? Otherwise, it could perhaps be omitted, since it doesn't add to the point of this manuscript without context.

Reply 2-4: We have added an introductory sentence for the 2016 episode in the intercession (Lines 75–76).

Comment 2-5: Lines 207-210: Papers by Sakaguchi et al. (2011) and Yamaguchi et al. (2011) analyzing core samples from IODP drilling of the Nankai frontal thrust both concluded that coseismic slip inducing frictional heating has affected very shallow regions in this zone (equivalent to along line D5, approximately). This is evidence that megathrust earthquake slip does propagate through the tremor zone, and should be mentioned here as well.

Reply 2-5: We thank the reviewer for this suggestion. The revised manuscript now clearly states the possibility of slip propagating through the tremor zone by referring to Sakaguchi et al. (2011) (Lines 281–283).

Comment 2-6: Figure 2. The black contours of tremor probability overlain on the rainbow bathymetry and heavy black fault traces in 2A are hard to clearly see and interpret. Perhaps a modified color scheme would be useful here. It's not clear to me that the color representation of bathymetry in A conveys much information that is not also shown in B as a grayscale, so maybe just use that palette for both, with color contours of tremor for A? Also, in 2B, the blue solid and dashed lines are not identified in the caption.

Reply 2-6: We have changed the contour color to blue for visibility. We have added an explanation for the purple dashed line in the figure caption (Lines 314–315).

Comment 2-7: Figure 4. The “slip-inhibitive” purple dashed line is described in the key as “on the megathrust” but is shown overlain on the frontal thrust, not the megathrust, except for in the panel labeled D7. This is a bit confusing, although I think that I understand what the authors intended, which is that slip transfer to the branching frontal

thrust inhibits the tendency for slip on the underlying, lower-dip megathrust (décollement). I suggest modifying this figure and its key to make a bit more clear what is being highlighted.

Reply 2-7: We have decided to remove the key saying “megathrust”, to avoid potential confusing.

Comment 2-8: Figure 5. The blue dashed hyperbola in panel C appeared at first look to be a set of blue dots, about the same size as the red and orange dots, at least in the provided pdf file. I could only see that it was actually a dashed line when I enlarged the figure to a very large size on screen. Perhaps the figure can be edited to more clearly distinguish the dashed line from the dots even at low magnification.

Response 2-8: We have increased the length of the line segment along the dashed line for better visibility.

Response to Reviewer #3:

Comment 3-1: In this well-written manuscript the authors present high-resolution tremor locations, bathymetry, and seismic reflection images that place new constraints on the up-dip limit of shallow slow slip earthquakes in the Nankai subduction zone. In most of the study region, the up-dip limit of tremor is spatially correlated with the intersection of the subduction décollement and the main frontal thrust suggesting that changes in fault geometry and/or associated changes in fluid pressure may act as a barrier to slow slip. This finding has important implications for the up-dip limit of megathrust earthquakes and tsunami generation I recommend publication of this manuscript in Nature Communications after considering my minor comments on the manuscript and figures below.

Reply 3-1: We thank the reviewer for the positive feedback.

Comment 3-2: Labelling of regions R1-R3 and profiles D1-D7 in text and figures -- I suggest modifying the labelling of the regions and profiles to make it easier for the reader

to understand which profiles correspond to which regions. Based on my understanding, profiles D1 and D2 occur in region R1; profiles D3, D4, and D5 occur in region R2; and profiles D6 and D7 occur in region R3. As one possible approach, if the regions were labelled A, B, and C, then the seven profiles could be labelled A1, A2, B1, B2, B3, C1, and C2 making it easy to see which profiles correspond to which regions.

Reply 3-2: In accordance with the reviewer's suggestion, we have changed the name of regions and profiles (Figures 2 and 3). Also, to include lines profile A2 into region A, we have slightly shifted the boundary between regions A and B eastward (Figure 2).

Comment 3-3: Line 95 – I suggest changing “crosscut” to “intersect”. From the information presented in this paper, it is not clear to me that the right-lateral strike-slip faults offset the decollement or are restricted to the upper plate.

Reply 3-3: We have changed it into “intersect” (Line 97).

Comment 3-4: Line 135 “Influence of geometry and pore fluid pressure variations on slip propagation” section -- In this section, I recommend adding a few sentences comparing the new results from Nankai presented in this manuscript with what has been discovered about the up-dip limit of shallow slow-slip earthquakes in other subduction zones, such as the Hikurangi, New Zealand (e.g., Wallace et al., 2016, Science) and Costa Rica (e.g., Davis et al., 2015, EPSL) subduction zones.

Reply 3-4: We agree with the importance of a comparison to other subduction zones. We notice that the slow earthquakes in Hikurangi and Costa Rica have potentially propagated up to the trench, but the spatial resolution of their source locations would not be sufficient to discuss structural controls on their updip limit. We have added a sentence for this discussion in the last section (Lines 292–295).

Comment 3-5: Lines 147-149 – I was not able to follow why the change in dip angle would increase the normal stress. I can understand how the change in geometry would result in a geometric barrier to slip and arguably the shear stress that would need to be overcome would increase from the decollement to the frontal thrust. But with the steeper

dip (and shallower depth) of the frontal thrust I would expect the normal stress on the faults to decrease toward the trench. I recommend the authors' clarify this point to the extent possible

Reply 3-5: Since the first principal stress axis is sub-horizontal, kink in the megathrust can increase the normal stress. We have added a detailed explanation for it, as another reviewer has a similar concern (Lines 159–171; Lines 423–446).

Comment 3-6: Paragraph starting at line 171 – I suggest adding a reference to Wallace et al. (2012, *Geology*) or similar paper that has proposed variations in fluid pressure as affecting the slip behavior of subduction plate interfaces. If high fluid pressures along the decollement are required for slow slip to occur, then it seems plausible that fluid could be leaking up the frontal thrust, reducing fluid pressures at the intersection with the decollement and therefore providing an up dip limit to slow slip.

Reply 3-6: We have renewed the corresponding paragraph regarding LVZ in response with another reviewer's comment (Lines 202–230). In the renewed paragraph, we have mentioned the possibility that the steep thrust could act as a fluid pathway that leaks fluid from the LVZ, which was suggested by the reviewer (Lines 221–222).

Comment 3-7: Line 184 – I suggest changing “overwrite” to “cut across” or “obscure”.

Reply 3-7: We have changed it into “obscure” (Line 233).

Comment 3-8: Line 211 – I agree with the authors that the geometrical barrier () identified in their study might prevent a future megathrust earthquake from reaching the trench, but it is also possible megathrust displacement along the decollement may track up the frontal thrust and still displace the seafloor.

Reply 3-8: We have reorganized the paragraph and mentioned that possibility (Lines 275–287).

Comment 3-9: Figure 2 – The tremor probability contours in Figure 2a cannot be deciphered at the scale of the figure. If this figure cannot be expanded in the main paper,

I recommend that only 1-2 key contours be shown and the expanded figure with more contours be included in the supplemental material.

Reply 3-9: We have changed the contour color and reduced the number of contours to enhance visibility.

Comment 3-10: Figure 2 and 5 – I recommend adding horizontal scales to these figures.

Reply 3-10: We have added horizontal scales to these figures.

Reply to reviewers' comments

Reviewer #3 (Remarks to the Author):

All of my concerns with the original manuscript have been appropriately addressed by the authors in the revised manuscript and I recommend publication in Nature Communications.

Reply: We thank the reviewer for the positive and encouraging comments.

Reviewer #4 (Remarks to the Author):

Dear Editor,

As was pointed by previous reviewers, the manuscript by Akuhara et al. presents an interesting correlation between a slow slip-tremors event in Nankai Trough and the geometrical features of the interplate boundary and continental wedge structures observed in high quality active seismic lines. This observation is important for the open research line associated to the interplay between seismotectonics and structural/geodynamical/geological evolution of subduction zones. As I'm participating in a second round of revision, I give some comments of the last version of the manuscript, considering only the points that, in my view, were not previously solved in the revision process. Below, I give comments after an R, referencing the corresponding lines in the text and the beginning of specific paragraphs.

Reply: We thank Reviewer #4 for joining the review process and for providing thoughtful and constructive comments. We appreciate the reviewer's recognition of the importance of our study and have carefully addressed all the points raised in the following responses.

Lines 75-76: "For comparison, the previous tremor episode...." R: Please include the location of this previous tremor episodes in a map figure. This should be important for interpretation.

Reply: We appreciate the reviewer's suggestion. While we understand the importance of showing the location of the 2016 tremor episode, we are concerned that adding these data

to Fig. 1 could cause unnecessary confusion due to the difference in the quality between the catalogues. Instead, we have presented the tremor distribution of the 2016 episode in Supplementary Fig. 9 and referred to this figure in a later section (Line 251), where the previous tremor locations are more relevant to the discussion.

Lines 92-93: "... outer wedge situated on the trenchward side of a megasplay fault...."

R: In Fig.2 it is clear that the correlation between the magasplay fault and the downdip of tremor events is even better than the correlation between the updip of this activity and the "kink" barrier. In my opinion profiles figures and text should be modified to include a more complete discussion of the downdip relation between tremors and continental wedge structure.

Reply: We agree with the reviewer's comment. In fact, Reviewer #2 in the previous round made a similar suggestion. Considering these comments seriously, we have substantially revised the manuscript to expand the discussion on the downdip limit of tremors (Lines 159–175, 262–293). We have also modified the manuscript title to emphasize both the downdip and updip limits.

Lines 93-95: "... The outer wedge is divided into two geological 93 zones..." R: Please include the limit between ITZ and FTZ in figure 2. This should help to note that the updip limit of slow earthquake is correlated with this limit only in the zone B.

Reply: We have added the ITZ–FTZ boundary to Fig. 2a, as suggested by the reviewer.

Lines 96-97: ..two subparallel right-lateral strike-slip faults located in the ITZ... R: Where are this right-lateral structures?. Please include the reference to the corresponding Figures.

Reply: We have added references to the corresponding figures as suggested (Line 110).

Lines 114-119: "...to region C, which lacks such features, aligns with the eastern boundary of the tremor distribution. Furthermore..." R: C is the estern limit of this tremor

episode, but this does not mean that future tremors episodes could not be located seaward in the segment C. To interpret that, the authors should analyze a much longer period of seismicity along the margin. In my view the analysis do not prove the tectonic control of the updip seismicity in C.

Reply: We appreciate the reviewer's insightful comment. Indeed, our analysis focuses only on the most recent tremor episode. Because the tremor distributions from earlier episodes are not well constrained, we decided not to extend our interpretation to previous or future episodes. Accordingly, we have deleted the relevant paragraph from the manuscript.

Lines 154-155:....current study indicate that the propagation of SSEs along the décollement is hindered at the kink or branching points..." R: Respecting to figure, the correlation between the updip of SSEs and the kink or branching points is clearly observed at the profiles B and in the profile A2. However, analyzing the profile A1 (in the supplementary material) the "kink" seems to be located landward from the point interpreted by the authors, then at the profile A1 the correlation with the updip of SSEs is not clear. Geometrically, the "kink" points seem to be correlated with the landward limit of the FTZ high. Maybe this observation is debatable, but in any case, it is important the inclusion of figures of the seismic profiles without interpretation (in the supplementary material) to facilitate the observation of reflectors near the décollement.

Reply: While we are confident in the location of the kink, which we define as the intersection between the décollement and the steep thrust for profile A1, we acknowledge that its interpretation may be debatable. Following the reviewer's suggestion, we have added the profiles without interpretation to Supplementary Figs. 5–7 to facilitate direct observation of the reflectors.

Lines 159-164: "...the steep thrust experiences higher normal stress than the gently dipping section of the décollement..." R: Respecting the discussion of Normal stresses, the authors include an analysis that is correct under the assumptions of the horizontality of the maximum stress and constant frictional conditions. However, both assumptions are difficult to associate with an accretionary wedge deformed under compressive forces,

gravitational forces and different internal and basal effective friction coefficients. In fact, the formation of a kink should be related precisely with a more complex stress field inside the wedge (see for instance the simple model of Dahlen et al. 1984, Noncohesive critical Coulomb wedges: An exact solution). On the other hand, the observation of the current and paleo deformation fronts should be related with variations of the basal effective friction coefficient and maybe with the aseismic deformation of sedimentary units near the deformation front. In my opinion, a more detailed discussion of the stress and strain field in accretionary wedges should be considered for the interpretation of structures and the SSEs episode.

Reply: We greatly appreciate this constructive comment. Following the reviewer's suggestion, we estimated the orientation of the maximum principal stress using the Coulomb wedge theory. In this calculation, reasonable parameters for the friction coefficient and related values were adopted (see Supplementary Table 1), based on laboratory experiments conducted for the study region (Okuda et al., 2021, *JGR*). Even when considering the tilted stress orientation estimated in this way, our interpretation—that the steep thrust experiences higher normal stress than the gently dipping section—remains valid (Lines 547–577). Furthermore, Okuda et al. (2020) reported a higher friction coefficient for the décollement beneath the FTZ than that beneath the ITZ, which would further increase fault strength along the steep thrust, consistent with our interpretation (Lines 188–190).

Lines 237-239: "...This lack of deformation is likely due to a stress shadow caused by the subduction of the Paleo-Zenisu ridge.." R: where is the Palo Zenisu_Ridge? Please indicate in maps and profiles. What is the definition of stress shadow used here? The authors of the reference 47 defined shadows zones seaward from subducted oceanic features and below inactive decollements, which seems to be not the case of profiles C. What is the subducted oceanic feature (observed in the profile) that generates the "stress shadow"? Please indicate in maps and figures.

Reply: We acknowledge that the previous version of the manuscript was somewhat unclear. We have now indicated the location of the subducted Paleo-Zenisu ridge in the profile (Supplementary Fig. 4). The map view of the Paleo-Zenisu ridge was already presented in previous figures (e.g., Fig. 1), but we have improved clarity by explicitly

labeling it as the “subducted Paleo–Zenuis ridge.”

The numerical setup of Miyakawa et al. (2022) may not fully represent the situation along profiles C1 and C2, as the height of the subducting ridge (or topographic high) in their model is much smaller (~100 m) than that in our study area. In this study, we consider that the stress-shadow zone (or zone of reduced tectonic loading) extends up to the shallow portion near the seafloor—at least to the depth where well-preserved horizontal strata are observed. This broader effect is consistent with the simulation results of Sun et al. (2020), who modeled a subducting seamount of comparable scale to that in our study. In addition, we found that the horizontal strata observed along profile C1 may be related to another undocumented ridge, separate from the Paleo–Zenuis ridge. We have clarified this interpretation in the main text (Lines 304–306) and in Supplementary Fig. 4.

Supplementary material: There are some signs “Error!” in figure captions.

Reply: All figure caption errors in the Supplementary Material have been corrected.

Reviewer #5 (Remarks to the Author):

Review by Madison Bombardier of the manuscript “Geometrical barrier determines the updip limit of slow earthquake slip” by Akuhara et al.

This manuscript uses a previously-published tremor catalogue to examine the relationship of shallow slow slip (using tremor as a proxy) to subduction zone structures imaged by seismic reflection in Nankai. Their main finding is the interpretation that the updip limit of slow slip is bounded by an abrupt steepening of the main thrust fault (top of decollement) in most of the region. They infer that the impedance of slow slip on the steep frontal thrust segment may imply impedance of coseismic slip during regular earthquakes, which may have consequences on tsunami generation and associated hazards.

Three reviewers during the first round of peer review requested: (1), clearer and more

robust mechanistic explanations of slip impedance along the steep frontal thrust in terms of pore-fluid pressure and stress; (2), discussion of the downdip limits of tremor; and (3), many editorial changes. The authors have reasonably addressed most of the comments from these reviewers, with some exceptions (see my comment number 29 below); however, some explanations and claims are vague or poorly supported.

The work presented in this manuscript is important to the understanding of subduction zone science, and represents an effective use of tremor catalogues for the study of this topic. After minor revision considering my comments below, this manuscript is worthy to be published. I have two main comments, and more than 3 dozen minor comments.

Reply: We thank Reviewer #5 for joining the review process and for providing thoughtful and constructive comments. We appreciate the reviewer's recognition of the importance of our study and have carefully addressed all the points raised in the following responses.

(1) Main comment:

The authors claim in multiple locations throughout the manuscript that the tremor catalogue they use contains “precise” tremor locations; however, evidence for this claim is not provided. The only relevant information provided is that the horizontal uncertainty on each tremor epicentre is estimated by one half the width of the 95% confidence interval to be “a few kilometres” (lines 72–73). It is not clear to me that this is precise, or even more precise than other methods, nor do I think that such a claim is necessary. The primary advantage of Bayesian inference is that solutions are provided as probability distributions, which gives unique 3D probability distributions of each tremor event from which uncertainty can be quantified; there is no guarantee of precision. In other words, this catalogue provides robust uncertainty estimates. Because of this, the tremor catalogue may be considered reliable or credible, but not precise. This is especially true in the context of this analysis where multi-channel seismic reflection data is used with resolution on the order of 100 m (line 82).

Despite that the primary strength of this tremor catalogue is the detailed uncertainty estimation, the authors do not provide quantitative estimates or distributions of these uncertainties, nor do they consider them in their analysis. Supplementary figure 1 shows

all tremor epicentres with 95% confidence interval error bars, but the superposition of the densely-clustered events obscures most of this information.

Here I make several suggestions to address these issues:

- remove descriptors such as “precise” everywhere related to the tremor catalogue
- in the main text, replace “a few kilometres” (lines 72–73) with mean or median horizontal uncertainty estimates, and/or state the range within which 50% or 75% or 90% (or whatever) of horizontal uncertainties are in.
- at the location where the tremor catalogue is introduced in the main text, briefly explain the advantage that Bayesian inference provides to this catalogue in terms of robust uncertainty estimation. Probabilistic solutions are used to represent the entire catalogue as one probability map, which provides more information than a scatterplot of point sources. A probability map is an unusual way to represent a catalogue of seismic events, and you should clearly explain the importance and novelty of this to encourage readers to adopt similar methods in future work.
- change supplementary figure 1 to better represent the geographic distribution of uncertainties. I suggest to segment the area into 1 km (or 5 km, or whatever) squared cells and colour each cell according to the average (or median) uncertainty of tremor events in that cell. In this case, you would need two panels in the figure: one for NS uncertainty and one for EW uncertainty. This method would allow the reader to view the heterogeneous geographic distribution of uncertainty over the study area. I expect average uncertainties vary systematically over the region, which may be useful information for the main analysis. In particular, I wonder about the EW uncertainties on the downdip tremor patch in the far north of the study area (region C). This type of information should be clearly discernible from this figure. An example can be found in the supplementary information of Bombardier et al 2025 (doi: 10.1016/j.tecto.2025.230752).

Reply: We appreciate the reviewer’s insightful comments. As suggested, we have removed subjective descriptors related to tremor locations, such as “precise,” throughout the manuscript and have emphasized the primary advantage of the method—its ability to provide reliable uncertainty estimates—in the Introduction section (Lines 73–78). Although this may differ slightly from the reviewer’s perspective, we have retained the phrase “which provide better constraints than conventional time-difference data,” as the

superiority of amplitude data over time-difference data has been demonstrated for this dataset (Akuhara et al., 2023, *GJI*).

In addition, following the reviewer's suggestion, we have added panels showing the median uncertainties within each geographic cell to Supplementary Fig. 1, and we now provide a more quantitative description of the uncertainty in the main text (Lines 81–83).

(2) Main comment:

Interpretations made in this analysis require a framework of SSEs wherein slow slip directly generates/triggers tremor, and wherein both tremor and slow slip occur on the megathrust fault (in this manuscript, the top of decollement). It is not my understanding that this framework is universally accepted as certain fact, thus the high degree of certainty with which it is presented is not warranted.

Regarding lines 60–62: it has been shown that tremor does indeed tend to occur near the slip front in Cascadia (eg Hall et al 2019, Bartlow et al 2011, Luo and Liu 2019), but I do not think there is sufficient evidence to use the word “triggered”, since there may be multiple processes occurring at the slip front that may cause, generate, or trigger tremor. In addition, shallow slow slip may not be perfectly analogous to deep slow slip where this has been studied more extensively. The brittle-asperity-in-ductile-matrix model (Luo and Liu 2019) is a plausible hypothesis (among others) to explain some elements of tremor and slow slip, and using this framework is justifiable, but it should be presented as a framework or base-level assumption rather than a known fact about the physical mechanisms.

Regarding lines 149–153: this reasoning is fuzzy. If you assume tremor is directly caused by slow slip, as the previous sentence states, how can the depth of tremor “most likely” be “similar” to that of SSEs? Would it not be certain that the depths are exactly the same? Reference 19 (Toh 2018) states that some VLFs likely occurred on a splay fault, which indicates that the depths of these VLFs are not localized entirely to the megathrust fault (insofar as depth constraints on VLFs support such an observation); or that the location/structure of megathrust fault itself is uncertain or complex. In addition, deep tremor in Cascadia represented as probability resulting from a Bayesian location method (Bombardier et al 2025) indicates that the depth of tremor can deviate significantly from

the top of the subducting oceanic crust and is inconsistent along the margin. Again, deep tremor/SSEs may not be so analogous to shallow tremor/SSEs, but given the difficulty of adequately constraining depths of tremor/LFEs/VLFs/slow slip, and the uncertainty regarding the relationship between these phenomena, consideration that these processes may occur beyond the megathrust fault is warranted. The possibility of tremor and SSE activity beyond the megathrust fault has important implications for discerning active structures and understanding hazards, and the study of this topic should not be discouraged by premature or simplifying assertions.

My suggestion to address this issue is to clearly explain that the analysis and interpretations presented in this manuscript requires the assumption of a specific framework/hypothesis about the nature of the relationship between tremor and slow slip, and about which structures host this activity due to the fact that tremor depths are not resolved in this catalogue. You already have many references to use to justify the use of such a framework. Also explain that variation exist between regions globally and uncertainty regarding the depth and host structures for such activity is prevalent, more work on these topics is needed. These changes largely involve reframing and rephrasing at various locations in the manuscript.

Reply: We thank the reviewer for this constructive and thoughtful comment. In the revised manuscript, we have clarified the working hypothesis in the Introduction section—that tremors accompany SSEs—and immediately followed it by noting the presence of observations that do not conform to this framework, in order to clarify its limitations (Lines 61–65). We agree that synchronization among tremors, VLFs, and SSEs has been most frequently reported in deep slow-earthquake regions (i.e., downdip of the locked zone). However, in the Kumano-nada region, synchronization between VLFs and SSEs has also been observed (Nakano et al., 2018; Edgington et al. 2025), owing to the high-sensitivity pressure gauges deployed by the DONET network (Lines 65 – 67).

In the Results section (Lines 319–350) , we have added a subsection titled “**Limitations and opportunities**” to discuss two key limitations not considered within the current framework: (1) the possibility of SSEs without associated tremors, and (2) the possibility that tremors occur on structures other than the décollement. For the latter, we refer to the recently published centroid moment tensor catalog of VLFs (Yamamoto et al., 2022), which shows focal depths and mechanisms indicating that most VLFs occur near the

décollement. Because tremors and VLFs are spatiotemporally synchronized (supplementary Fig. 10), we infer that tremors also occur near the décollement, possibly within a mélange shear zone embedded in the LVZ. This interpretation has been explicitly stated in the new subsection.

Overall, we have designed the added section not only to acknowledge the limitations of the current framework but also to highlight potential future research directions. We firmly believe that this addition strengthens the manuscript by clarifying the scope of our interpretation while pointing toward avenues for further investigation.

(3) The abstract should include why your results matter. I suggest simplifying the summary of results in the abstract to make space for the broader interpretations/impact of the results mentioned in the “introduction” and “impacts” sections.

Reply: We have revised and reorganized the abstract accordingly to better emphasize the broader implications and significance of our results.

(4) Lines 51–53 it may be relevant to the present manuscript that the forearc mantle corner has also been shown as a boundary to slow slip by Sawaki et al (2021) and Bombardier et al (2024).

Reply: While we recognize the importance of these studies, we decided not to include them because they focus on deep slow earthquakes, which are beyond the scope of the present study.

(5) Lines 99–100 regarding a new deformation front, this should have a citation.

Reply: We have added the appropriate citations to support this statement (Line 113).

(6) Line 101 the tremor probability map is mentioned for the first time without explanation of what it is. A reference to the methods section where the probability map is mathematically defined is missing.

Reply: We have added a reference in the Introduction to the Methods section that defines the tremor probability map (Line 78).

(7) The ITZ-FTZ boundary denoted in Figure 2b by the purple dashed line would be helpful in panel (a) of this figure, so that the spatial correspondence between this boundary and the updip limit of tremor is very clear.

Reply: We have added the ITZ–FTZ boundary to Fig. 2a, as suggested by the reviewer.

(8) It is not that easy to discern the annotations in Figure 2b. The dark colour of the bathymetry shading makes it difficult to clearly see the thin black font of the text and transparent blue/red lines. It would be better to make these annotations more pronounced, possibly by decreasing their transparency, making the text font bolder, and/or increasing the transparency of the grey bathymetry shading.

Reply: We have adjusted the background color of Fig. 2b to a lighter tone so that all annotations are more clearly visible.

(9) Lines 115–117 here is an example where it is important to know the uncertainties on tremor in region C. “much further landward” may not be a well-supported observation depending on the uncertainties.

Reply: We have added a new panel showing the median uncertainty within each geographic cell to Supplementary Fig. 1. This figure confirms that tremor occurrence extends farther landward in region C.

(10) Lines 118–121 This sentence is long and awkward, and should be in present tense rather than in a past tense narrative-style description of what you did. For example, try “To explore the apparent agreement between the tremor distribution and bathymetric features, we [analyze]...”

Reply: We have rephrased the sentence to “To further explore the correspondence between the tremor distribution and bathymetric features” and changed it to the present

tense (Lines 132–134).

(11) Supplementary figures 2–4 have a “Error! Reference not found” note in the captions. Also the caption in supplementary figure 2 should reference “inverted triangles” (plural), not “inverted triangle”. These inverted triangles in profile A1 are difficult to see clearly; I did not realize they were triangles at first. Perhaps make them bigger with a black outline?

Reply: We have corrected the erroneous references to Fig. 3 and changed the color of the inverted triangles to a darker tone to improve their visibility.

(12) Lines 124–125 “almost subparallel” doesn’t really make sense. I suggest rephrasing as “almost parallel” or simply “subparallel”. In addition, this short sentence could be joined with the previous to make it less awkward. For example, “The most prominent features ... oceanic crust and decollement, which are subparallel within the ITZ.”

Reply: We have changed “almost subparallel” to “subparallel” and combined the two sentences, following the reviewer’s suggestion (Line 137).

(13) Line 131 I suggest changing “new” and “old” deformation front to “deformation front” and “mega splay”. To say “new” implies it is a new interpretation in this manuscript, and it can confuse the reader when there is no discussion of this new feature. This requires changes to annotations in figures as well.

Reply: We agree that the terms “new” and “old” could be misleading. We have replaced them with “younger” and “older” to better reflect their relative development stages while avoiding the implication of a new interpretation (e.g., Line 113). We did not adopt the term “megaspaly” suggested by the reviewer, because in this region the term is usually used for the fault along the boundary between the inner and outer wedges. The figure annotations have been updated accordingly.

(14) Lines 143–144 Supplementary figure 4 does not actually show the tremor

distribution along profile C1 and C2. The map in Figure 3 is a more appropriate reference for this observation. However, as a reader, I find that I need to refer to both figure 3 and supplementary figure 4 to fully understand this observation, which is cumbersome. It would help to include the tremor probability distribution along each profile in supplementary figures 2–4 (as is done in figure 3).

Reply: We have added the tremor probability distributions to the profiles in Supplementary Figs. 2–7, as suggested. This addition allows readers to understand the observations without referring to both Figure 3 and the supplementary figures.

(15) I suggest including an annotation in supplementary figure 4 to identify the subducting ridge. I can guess what constitutes the ridge, but better to identify it clearly because I may be wrong.

Reply: We have added annotations indicating the location of the subducted ridge in Supplementary Fig. 4, so that readers can clearly identify it.

(16) Lines 153–155 this is the only location figure 4 is referenced. Perhaps this early reference to this figure should be removed so the figures can be ordered logically; I do not think it adds much at this point in the manuscript. Figure 4 represents a summary of your interpretations, so it should be explained near the end of the manuscript and used to support such a summary. Notably, a brief summary of the new observations and interpretations do not really exist, but are apt.

Reply: We appreciate this helpful suggestion. We have moved the figure to a newly added subsection titled “Conclusions” (Lines 383 – 414). In this section, each finding is summarized in bullet-point form, and the corresponding portions are displayed in the figure with matching numbering for clarity. We firmly believe that this revision better demonstrates the necessity and relevance of the figure than in the previous version.

I think lines 153–155 should say “... the propagation of SSEs along dip is hindered...” since figure 4 also shows the ridge stress shadow and the strike slip faults hinder along-strike SSE propagation.

Reply: We have added “along-dip” to this sentence (Line 181).

(17) Lines 155–158 reference 40 (Wech and Bartlow 2014) is a study from Cascadia that observed one instance of an SSE occurring without tremor for a short time. While I agree that these cases seem infrequent in Cascadia, I do not think this is relevant to shallow SSEs in Nankai. It is reasonable to exclude SSEs without tremor, but this should be explained and justified near line 60 where the framework, assumptions, and unknowns are explained. And this choice should be supported with observations for the relevant region. Also, related to my main comment (2), SSEs occurring without tremor are incompatible within a framework wherein slip generates tremor, as stated just three sentences before this. If there are SSEs that occur without tremor, such that they are excluded from this analysis, maybe the framework needs to be revised?

Reply: As mentioned in our response to comment (2), the limitation of our framework—particularly the possibility of SSEs occurring without accompanying tremors—is now discussed in a newly added subsection (Lines 319–350). We also briefly note this limitation in the Introduction (Lines 64–65) and at the beginning of the discussion section (Lines 182–184).

(18) Line 175 “distinctly different” is a redundant phrase. “Distinct” and “different” are synonyms.

Reply: We have removed “distinctly”, as suggested (Line 205).

(19) Line 176 rephrase this as “A stratified pattern observed at the trench side of the steep thrust is absent on the landward side.” and it would be better to reference figure 3 rather than supplementary figure 2, since this feature can be seen in the main-text figure.

Reply: We have revised the sentence as suggested (Lines 205–206) and now refer to Figure 3 instead of Supplementary Figure 2.

(20) Line 185 “in distance” is unnecessary because kilometre is a unit of distance.

Reply: We have removed “in distance”, as suggested (Line 242).

(21) Line 186 briefly explain why the slip is diffusive, and why diffusive slip is characterized by a hyperbola.

Reply: We acknowledge that the term “hyperbola” was erroneously used and have corrected it to “parabola,” where the propagation distance is proportional to the square root of time. This parabolic feature commonly appears in diffusion processes. Thus, the term “diffusive” is used here in a descriptive sense, rather than implying a specific physical mechanism. We have added an explanation in the main text that this parabolic pattern can be interpreted as the result of an SSE propagating along a heterogeneous fault surface, where tremors occur on small-scale brittle patches (Ando et al., 2012) (Lines 244–246).

(22) Line 191 “bounded” is different from “impeded”. In this manuscript, the tremor migration is impeded, not bounded by the strike slip faults.

Reply: We have rephrased this sentence to “Tremor activity in April 2016 also appears to have terminated when the migration front reached this strike-slip fault” to more accurately convey our intention (Lines 250–251).

(23) Line 197 “horizontally” is unnecessary.

Reply: We have removed it, as suggested (Line 256).

(24) Lines 197–198 could slip have propagated along one of the strike slip faults? This could explain why the second small SSE initiated landward of the termination of the larger SSE (roughly along the strike of the strike slip faults).

Reply: This is an interesting possibility, but assessing whether slip propagated along the strike-slip faults is beyond the scope of this study, as it would require detailed evaluation of the errors and the definition of the projection profiles.

(25) grey dots in figure 5b are not explained anywhere.

Reply: We have added the explanation to the figure caption.

(26) Line 203 rephrase to “Although its spatial distribution is not well constrained, the potential role of pore-fluid pressure in the study region is worth discussing here.”

Reply: We have rephrased it, as suggested (Lines 214–215).

(27) Lines 205–206 the transparent facies are observed immediately above the decollement, not oceanic crust, according to figure 3.

Reply: We have changed it to “above the décollement” (Line 217).

(28) Line 211 rephrase to “... high pore-fluid pressure is interpreted to be sustained by ...”

Reply: We have rephrased it, as suggested (Lines 222–223).

(29) Lines 212–216 this section was added in response to comment 1-15 from the first review. It is awkward and oddly written as a string of short facts that are not presented cohesively. How exactly are these inferences relevant to the analysis? Explain “tectonic push” and what “these structures” are that may create tectonic push; does this refer to basement highs? Not clear.

Reply: We agree that this paragraph was not cohesively written and only loosely related to the main discussion. As it was an overreaction to Comment 1-15 from the first review, we have decided to remove it in the revision.

(30) Lines 237–238 it is my understanding that subducting ridges and seamounts lead to highly fractured/deformed accretionary prisms/forearcs. In order for a lack of

deformation in the prism to result from the stress shadow of a subducted ridge, it seems to me that the undeformed prism material would have to have been accreted after the ridge (in its shadow) in order to remain relatively undeformed. If the prism material was accreted first, then the ridge would have subducted under it, and the explanation given in the manuscript does not make sense to me. In this case, lack of deformation within the prism would not be a reliable indicator of the existence of a stress shadow. Please clarify the accretionary history and the relationship between subducted ridges/seamounts and prism deformation. Maybe there is an alternative explanation for the reflective strata?

Reply: We appreciate the reviewer's thoughtful comment. According to Kimura et al. (2011), the ITZ had already formed at the initiation of ridge subduction. Nevertheless, even if the ridge subducted later, the tectonic loading subsequently applied to the prism could have been reduced by the stress shadow of the subducted ridge, as demonstrated by numerical simulations (e.g., Sun et al., 2020). This stress-shadow effect provides a reasonable explanation for the relatively weak deformation observed in this region compared to adjacent areas.

We do not mean that the overriding prism does not deform at all; rather, the deformation may occur only in a limited portion, as indicated by several sparsely distributed thrusts observed in the profile.

(31) Line 252 rephrase to "... large earthquake may have facilitated tremor ..."

Reply: The paragraph containing this sentence has been removed from the revised manuscript in accordance with another reviewer's suggestion.

(32) Line 257 I believe the boundary between the inner and outer wedges is the megasplay fault shown in figure 2b and mentioned on line 93. I suggest including the "megasplay" term here with reference to figure 2b so it is clear.

Reply: We have explicitly indicated the location of the boundary between the inner and outer wedges in Figure 2b, which indeed corresponds to the megasplay fault.

(33) Line 258 "distinctly differ" is redundant

Reply: The paragraph containing this sentence has been revised in accordance with another reviewer's suggestion, and the corresponding phrase is no longer used in the manuscript.

(34) Lines 262–263 please explain how a discontinuous stress state in the outer forearc wedge would affect slip on the megathrust.

Reply: The revised text now elaborates on how an abrupt change in the stress state can impede slip propagation along the megathrust (Lines 287–289).

(35) Line 264 the use of the term “separate” is confusing here. The tremor episode that is the focus of this manuscript occurs in the transient slip zone, so how can it separate the transient slip zone from the fully locked zone? I think this should be phrased “the downdip limit of this tremor episode may not be a universal representation, as SSEs and tremors have occurred further downdip in earlier episodes.”

Reply: We have revised the phrase to “the downdip limit for this episode may not necessarily correspond to the boundary between the transient-slip and locked zones” (Lines 290–291).

(36) Line 283 “numerous numerical” is a slightly awkward phrase. I suggest “multiple”, “several”, or any other synonym for “numerous”.

Reply: We have replaced “numerous” with “multiple,” as suggested (Line 367).

(37) Line 294 I do not think Hikurangi is a relevant example here because tremor is not closely associated in space or time with shallow SSEs (eg Todd et al, 2018). As such, the tremor location methods used in this paper would be useful for locating tremor in Hikurangi, but not be useful for approximating slow slip.

Reply: We have removed the reference to Hikurangi from this sentence, as suggested (Line 379).

(38) There are numerous grammatical errors and typos throughout the main text and supplementary material, in addition to the ones mentioned above. Text added in response to the earlier reviewers could be integrated better so that the text flows well and seems cohesive.

Reply: We have carefully checked the entire manuscript for grammatical errors and typos. In addition, rather than addressing each reviewer's comment in isolation, we have reorganized the text to improve the overall flow and coherence of the manuscript.

(39) Inconsistent use of "Figure" and "Fig" (eg line 212), and bolded text for figure references (eg line 206).

Reply: We have made the usage of "Figure" and "Fig." consistent throughout the manuscript and removed bold formatting from figure references.